# Arctic Ocean Simulations in the CMIP6 Ocean Model Intercomparison Project (OMIP)

Qi Shu[1,2], Qiang Wang[3], Chuncheng Guo[4], Zhenya Song[1,2], Shizhu Wang[1,2], Yan He[1,2], Fangli Qiao[1,2]

[1]First Institute of Oceanography, and Key Laboratory of Marine Science and Numerical Modeling, Ministry of Natural Resources, Qingdao, 266061, China
[2]Shandong Key Laboratory of Marine Science and Numerical Modeling, Qingdao, 266061, China
[3]Alfred Wegener Institute Helmholtz Centre for Polar and Marine Research (AWI), Bremerhaven, 27570, Germany
[4]NORCE Norwegian Research Centre, Bjerknes Centre for Climate Research, Bergen, 5007, Norway

*Correspondence to*: Qi Shu (shuqi@fio.org.cn)

**Abstract.** Arctic Ocean simulations in 19 global ocean-sea ice models participating in the Ocean Model Intercomparison Project (OMIP) of the CMIP6 are evaluated in this paper. Our findings show no significant improvements in Arctic Ocean simulations from the previous Coordinated Ocean-ice Reference Experiments phase II (CORE-II) to the current OMIP. Large model biases and inter-model spread exist in the simulated mean state of the halocline and Atlantic Water layer in the OMIP models. Most of the OMIP models suffer from too thick and deep Atlantic Water layer, too deep halocline base, and large fresh biases in the halocline. The OMIP models qualitatively agree on the variability and change of the Arctic Ocean freshwater content, sea surface height, stratification and volume, heat and freshwater transports through the Arctic Ocean gateways. They can reproduce the changes in the gateways transports observed in the early 21[st] century, with the exception of the Bering Strait. We also found that the OMIP models employing the NEMO ocean model simulate relatively larger volume and heat transports through the Barents Sea Opening. Overall, the performance of the Arctic Ocean simulations is similar between the CORE2-forced OMIP-1 and JRA55-do-forced OMIP-2 experiments.

## 1 Introduction

As the northernmost ocean on Earth, the Arctic Ocean is of great concern to researchers and the general public, particularly against the backdrop of the global warming. The decline of Arctic sea ice and warming of the water in the Arctic Ocean indicate that the Arctic Ocean has been experiencing rapid climate change (Onarheim et al., 2018; Stroeve and Notz, 2018; Polyakov et al., 2012, 2017; Danielson et al., 2020). The Arctic Ocean is projected to warm faster than the global ocean average, a phenomenon called Arctic Ocean amplification (Shu et al., 2022). The changes of the Arctic Ocean potentially affect the climate system beyond the Arctic. For example, the sea ice decline in the Arctic tends to cause cold winters and extreme weather events over the mid-latitude continents in the Northern Hemisphere (Li et al., 2020; Outten and Esau, 2012; Kim et al., 2014; Cohen et al., 2020), and the storage and release of freshwater from the Arctic Ocean can influence the large

scale ocean circulation by freshening the upper North Atlantic ocean (Jungclaus et al., 2005; Goosse et al., 1997; Wadley and Bigg, 2002; Shu et al., 2017; Zhang et al., 2021).

The Arctic Ocean is surrounded by the Eurasian and the North American continents (Fig. 1), which is connected to the Atlantic Ocean on both sides of Greenland and to the Pacific Ocean through the Bering Strait. The general circulation in the Arctic Ocean is the superposition of Atlantic Water flowing into and around the Arctic Basin and two main wind-driven

circulation features of the interior stratified Arctic Ocean: the Transpolar Drift Stream and the Beaufort Gyre (Fig. 1) (Timmermans and Marshall, 2020; Wang and Danilov, 2022). Warm and saline Atlantic Water enters the Arctic Ocean with two branches. One branch passes the Fram Strait and supplies the warm Atlantic Water layer of the Arctic Ocean (Våge et al., 2016; Beszczynska-Möller et al., 2012; Rudels et al., 2015). The other enters the Barents Sea and then Kara Sea and finally flows to the surface, intermediate and deeper layers of the Arctic Ocean (Karcher, 2002; Schauer et al., 2002;

Maslowski et al., 2004). The Atlantic Water circulates mainly cyclonically along the peripheries of the Arctic basins and is aligned with the continental slope (Karcher et al., 2003; Timmermans and Marshall, 2020). Fresh Pacific Water flows into the Arctic Ocean through the Bering Strait (Woodgate, 2018b; Woodgate and Peralta-Ferriz, 2021), and leaves the Arctic via the Fram Strait and Canadian Arctic Archipelago (Hu et al., 2019; Steele et al., 2004; Lique et al., 2010; Wang et al., 2021).

The Arctic Ocean is a large freshwater reservoir, which plays a critical role in the global climate system. Ekman convergence

associated with the Arctic atmospheric anticyclonic circulation centered in the Beaufort Gyre region leads to high liquid freshwater content in the Arctic Ocean (Proshutinsky et al., 2009, 2002). The Arctic Ocean receives large amounts of fresh water from river runoff, oceanic freshwater flux through Bering Strait, and net precipitation, and releases fresh water through Davis Strait and Fram Strait (Serreze et al., 2006; Dickson et al., 2007; Ilicak et al., 2016). Observations show that liquid freshwater stored in the Arctic Ocean has increased since the mid-1990s and stabilized in the 2010s with an unprecedented

amount of freshwater accumulated in the Amerasian Basin (Rabe et al., 2014; Polyakov et al., 2013; Wang et al., 2019b; Wang and Danilov, 2022; Solomon et al., 2021).

Water masses in the Arctic Ocean can be distinguished with five separate layers (Rudels, 2009), including a ~50-m-thick upper polar mixed layer, a 100–250-m-thick halocline layer, a 400–700-m-thick Atlantic Water layer, an intermediate layer below the Atlantic Water layer, and a bottom-most layer containing deep and bottom waters. Since the 1990s significant

warming signals have been observed in the upper polar mixed layer and Atlantic Water layer (Polyakov et al., 2012, 2020a; Ingvaldsen et al., 2021; Li et al., 2022; Steele et al., 2008), together with a weakening of the cold halocline layer and an increase of upward oceanic heat flux from the Atlantic Water layer in the eastern Arctic Ocean (Polyakov et al., 2020c).

Ocean/sea-ice and climate models are often used for scientific studies on the Arctic Ocean due to the limited amount of observational data under the harsh environmental conditions. To validate and further improve model performances in the

Arctic Ocean, the Arctic Ocean Model Intercomparison Project (AOMIP) and Forum for Arctic Modeling and Observational Synthesis (FAMOS) were initiated in 1999 and 2013 (Proshutinsky et al., 2001, 2016), respectively. Significant progresses have been made in AOMIP and FAMOS, for example, some systematic biases in Arctic Ocean models (such as a wrong circulation direction and an overestimation of the thickness and depth of the Atlantic Water layer) have been identified and

solutions to some of the issues have been recommended (Holloway et al., 2007; Golubeva and Platov, 2007; Zhang and Steele, 2007; Proshutinsky et al., 2007; Aksenov et al., 2016; Hu et al., 2019).

Focusing on the global scale ocean simulations, the Coordinated Ocean-ice Reference Experiments (COREs) and subsequent Ocean Model Intercomparison Project (OMIP) proposed by the WCRP (World Climate Research Programme)/CLIVAR (Climate and Ocean: Variability, Predictability and Change) Working Group on Ocean Model Development were also successively initiated (Griffies et al., 2009, 2012, 2016). COREs and OMIP aim to provide a framework for evaluating, understanding, and improving the ocean and sea-ice components of global climate and Earth system models contributing to the Coupled Model Intercomparison Project (CMIP). OMIP is an endorsed project in CMIP Phase 6 (CMIP6). The performances of COREs phase II (CORE-II) models in simulating Arctic sea ice, liquid freshwater, hydrography, and volume/heat/freshwater fluxes were comprehensively assessed (Wang et al., 2016a, b; Ilicak et al., 2016). Arctic sea ice simulations by the recent OMIP models were also evaluated by Tsujino et al. (2020). However, Arctic Ocean simulations by the OMIP ocean models, most of which were used as the ocean components of fully-coupled models in CMIP6, have not been evaluated systematically in model intercomparison studies.

In this work we analyse the Arctic Ocean properties simulated by the models participating in the CMIP6 OMIP to evaluate the latest ocean components of CMIP6 climate models in the Arctic Ocean. This work is to some extent an update of Wang et al. (2016a, b) and Ilicak et al. (2016), with one of the aims to examine if some progresses in simulating the Arctic Ocean have been made in the global ocean and sea ice models, from the previous CMIP5 phase to the most recent CMIP6 phase.

## 2 Methods

Most of OMIP models were used as the ocean components of the CMIP6 fully-coupled models; different from the latter, global ocean-sea ice models participating in the CMIP6 OMIP are driven by the specified common atmosphere forcing data sets. Their initial conditions of temperature and salinity are observational-based climatology from World Ocean Atlas 2013 (Locarnini et al., 2013; Zweng et al., 2013). Two versions of OMIP experiments were proposed in the framework of CMIP6, the version driven by the CORE2 forcing (OMIP-1) and the version driven by the JRA55-do forcing (OMIP-2). The CORE2 forcing contains the inter-annually varying atmospheric forcing and river runoff during 1948 to 2009 (Large and Yeager, 2009), which have been developed from the National Centers for Environmental Prediction/National Center for Atmospheric Research (NCEP/NCAR) reanalysis and the observation-based runoff data provided by Dai and Trenberth (2002) and Dai et al. (2009). The forcing is the same as that in the CORE-II project. The temporal frequency and horizontal resolution in CORE2 forcing are 6h and 1.875 °, respectively. The JRA55-do forcing used in OMIP-2 (Tsujino et al., 2018) is based on the Japanese Reanalysis (JRA-55) product from Kobayashi et al. (2015). For OMIP-2 simulations, JRA55-do forcing covers 1958 to 2018. It has higher temporal frequency (3h) and refined horizontal resolution (0.5625 °) compared to CORE2 forcing.

In this study, the Arctic Ocean simulations from 19 ocean-sea ice models (Table 1) participating in OMIP-1 and/or OMIP-2 experiments are evaluated. Eight models, including AWI-CM-1-1-LR, CESM2, CMCC-CM2-SR5, EC-Earth3, FGOALS-f3-

L, MIROC6, MRI-ESM2-0, and NorESM2-LM, provide both OMIP-1 and OMIP-2 simulations, which offer an opportunity to study the simulation differences related to forcing. Based on the OMIP protocol, each model should run for no less than five cycles of the forcing periods (1948–2009 for OMIP-1 and 1958–2018 for OMIP-2) to provide a model state that does not drift much anymore. Upon reaching the end of the year 2009 in OMIP-1 and 2018 in OMIP-2, the forcing is changed to that in 1948 in OMIP-1 and 1958 in OMIP-2. Here we mainly focus on the last cycle of each model to study the model performances in the Arctic Ocean.

The Arctic Ocean in this study refers to the Arctic Basin (Eurasian and Amerasian basins) and its surrounding shelf seas, including the Barents, Kara, Laptev, East Siberian, Chukchi, and Beaufort seas, as well as Baffin Bay (Fig. 1). Using this definition, the four gateways, including Bering Strait, Fram Strait, Barents Sea Opening (BSO), and Davis Strait (Fig. 1), can be conveniently used to study the volume, heat and freshwater exchanges between the Arctic Ocean and lower-latitude oceans.

This paper is organized as follows. Section 3 shows evaluation results, including the evaluation of mean hydrography, liquid freshwater content, changes in Atlantic Water layer, stratification, sea surface height, and gateway transports. Section 4 provides discussion and conclusions.

## 3 Results

We mainly evaluate OMIP simulated climatology and changes of Arctic Ocean hydrography, liquid freshwater content, Atlantic Water layer temperature, mixed layer depth, cold halocline base depth, sea surface height, and gateway transports, using the Polar science center Hydrographic Climatology (PHC3.0) (Steele et al., 2001), altimetry measurements (Armitage et al., 2016), and other relevant observations from published literatures. One has to keep in mind that although the datasets used to evaluate models are mostly based on observations, they also have biases and uncertainties.

### 3.1 Mean Hydrography

The PHC3.0 climatology is mainly based on observations before 2000, so the period 1971–2000 was chosen to evaluate the mean state in the models. The PHC3.0 climatology shows that the Arctic basin-mean temperature is cold in the surface and deep/bottom layers, and relatively warm in the Atlantic Water layer (Fig. 2). The Atlantic Water layer is mainly located at the depths between 150–900 m with a warm core between 200–400 m and 400–600 m in the Eurasian Basin and Amerasian Basin, respectively. Most models in OMIP-1 and OMIP-2 can reproduce the vertical structures of the temperature but exhibit large biases and inter-model spreads (Fig. 2). Several models, such as CanESM5, CMCC-CM2-SR5, CMCC-ESM2, and NorESM2-LM, cannot reproduce the warm layer, while the warm cores in CAS-ESM2-0, CESM2, FIO-ESM-2-0, and GFDL-CM4 are too warm. AWI-CM-1-1-LR has overall the best performance in representing the vertical temperature profiles in OMIP-1, although its warm core is biased warm in the OMIP-2 simulation. The more realistic simulations in

AWI-CM-1-1-LR may have benefited from its relatively high resolution (~24 km) in the Arctic Ocean, while the typical resolution of OMIP models is ~50 km.

Figure 3a–c shows the PHC3.0 and the multi-model mean potential temperature at 400 m, a depth that was used in previous model intercomparison studies (Ilicak et al., 2016) , which is close to the core of the warm Atlantic Water layer in the Arctic Basin. The eight models providing results for both OMIP-1 and OMIP-2 are used in the calculation of the multi-model mean here. The Atlantic Water layer temperature decreases along the pathways of the Atlantic Water. It is around 2 ℃ near the Fram Strait, and decreases gradually to about 0.8 ℃ near the Lomonosov Ridge and to about 0.4 ℃ in the Canada Basin (Fig. 4a). However, the Atlantic Water layer in both the OMIP-1 and OMIP-2 multi-model mean results is too cold in the Arctic Basin and too warm in the Greenland Sea and Norwegian Sea, and its warm temperature signal disappears rapidly along the propagation pathway in the Arctic Basin compared to the observations (Fig. 3b–e), which is similar to that in the CORE-II simulations (Ilicak et al., 2016). The warm biases in the Greenland Sea are smaller in relatively high resolution models, including AWI-CM-1-1-LR and CMCC-CM2-HR4 (not shown). Cold biases in the Atlantic Water layer in the Arctic Basin are also found in CMIP6 fully-coupled climate models (Khosravi et al., 2022; Heuzé et al., 2023). The cold biases in the Arctic Basin in OMIP may be caused by too weak heat transport through the Fram Strait and too cold temperature in the outflow from the Eurasian continental shelf through the St. Anna Trough (the same reason for the cold biases in CORE-II models as suggested in Ilicak et al., 2016). Figures 2, 3d and 3e show that the magnitudes of the Atlantic Water layer temperature biases are slightly larger in OMIP-1 than OMIP-2.

Another bias in the OMIP models is the too thick Atlantic Water layer shown by Fig. 4. The observed Atlantic Water layer is mainly located at 150–900 m depths, while the simulated Atlantic Water layer lower boundary is deeper than 1200 m (Fig. 4b and c). The depth of the highest temperature is also deeper in the models. The too thick and deep Atlantic Water layer, which is also found in AOMIP, CORE-II, and CMIP6 simulations (Holloway et al., 2007; Ilicak et al., 2016; Khosravi et al., 2022; Heuzé et al., 2023), is likely the consequence of too much spurious diapycnal mixing associated with the advection operator in low resolution models (Zhang and Steele, 2007; Holloway et al., 2007, Wang et al., 2018). Figures S1 and S2 indicate that the issue of too thick and deep Atlantic Water layer is common in OMIP models, and some models even did not simulate a warm Atlantic Water layer, such as CMCC-CM2-SR5, CMCC-ESM2, CanESM5-CanOE, CanESM5, and NorESM2-LM.

The biases of salinity profile in the Eurasian and Amerasian basins are shown in Fig. 5. For the upper polar mixed layer, both positive and negative biases are found in both the suites of OMIP models, and the multi-model mean biases are positive in both the Eurasian and Amerasian Basins in OMIP-1, but is negative in the Amerasian Basin in OMIP-2. The surface salinity biases in OMIP simulations are similar to that in CORE-II simulations (Ilicak et al., 2016), and are much smaller than those in CMIP5 and CMIP6 simulations (Shu et al., 2019; Khosravi et al., 2022). This may be related to the sea surface salinity restoring commonly used in standalone ocean-sea ice models (Wang et al., 2018), which is intended to sustain model stability and to avoid unbounded local salinity trends that can occur in response to inaccuracies in, for example, precipitation forcing (Griffies et al., 2009).

Large salinity biases are found in the upper 400-m ocean, and most OMIP models suffer from too fresh biases at subsurface (50–400 m) (Figs. 3f–j and 5). The maximum biases in OMIP-1 multi-model mean are −0.42 and −0.57 psu in the Eurasian and Amerasian basins, respectively. OMIP-2 has relatively larger biases with the largest biases of −0.49 and −0.77 psu in the Eurasian and Amerasian basins, respectively. Similar fresh biases are also found in CORE-II, and coupled CMIP5 and CMIP6 simulations (Ilicak et al., 2016; Shu et al., 2019; Khosravi et al., 2022). It is possibly caused by the absence of proper subgrid parameterization of brine rejection in these models (Ilicak et al., 2016; Nguyen et al., 2009). Different from most models, EC-Earth3 and IPSL-CM6A-LR have obvious positive salinity biases (larger than 1 psu in the Eurasian Basin) at subsurface, which has not been found in CORE-II simulations. In the Greenland Sea and Norwegian Sea, the biases of multi-model mean salinity are positive (Fig. 3i and 3j).

Both the simulated temperature and salinity show that OMIP models have large inter-model spreads, including the halocline and Atlantic Water layer (Figs. 2, 5, S1 and S2), which have also been found in AOMIP and CORE-II simulations (Holloway et al., 2007; Ilicak et al., 2016). The eight models providing results for both OMIP-1 and OMIP-2 indicate that OMIP-2 simulations have slightly smaller inter-model spread than OMIP-1 simulations (Fig. 6). The overall temperature and salinity biases and inter-model spreads in OMIP simulations are similar to those in CORE-II simulations. It indicates that there are no significant improvements in simulating the mean hydrography of the Arctic Ocean from CORE-II (CMIP5 phase) to OMIP (CMIP6 phase).

**3.2 Liquid Freshwater Content**

Liquid freshwater in the Arctic Ocean has strong implications on Arctic physical and biogeochemical environment and large-scale ocean circulation in the North Atlantic (Coupel et al., 2015; Ardyna and Arrigo, 2020; Zhang et al., 2021). To evaluate the liquid freshwater simulations, we use 34.8 psu, the mean Arctic salinity (Aagaard and Carmack, 1989), as the reference salinity to calculate liquid freshwater column and freshwater transport, which was commonly used in previous studies (Jahn et al., 2012; Wang et al., 2016b; Serreze et al., 2006; Haine et al., 2015). The liquid freshwater column (in meters) is calculated as follows:

$$\text{FWC} = \int_{-H_{ref}}^{0} (1 - S(z)/S_{ref})dz, \tag{1}$$

where $S(z)$ is salinity at depth z, $S_{ref}$ is the reference salinity, and $H_{ref}$ is the depth where seawater salinity is equal to the reference salinity. Integrating the freshwater column in an area one gets the volumetric freshwater content.

Observations show that liquid freshwater column in the Arctic Ocean is the highest in the Beaufort Gyre because of the Ekman convergence associated with the atmospheric Beaufort High (Fig. 7a). Both OMIP-1 and OMIP-2 models can reproduce this spatial pattern (Fig. 7). Consistent with the fresher biases shown in Figs. 5 and 6, OMIP-2 models simulated more freshwater content in the Arctic Ocean than the observations and OMIP-1 (Figs. 7 and 8). Freshwater storage in the Arctic Basin (where bottom topography is deeper than 500 m) is $55.8 \times 10^3$ km³ based on PHC3.0 climatology, and is $60.5 \times 10^3$ and $66.3 \times 10^3$ km³ in OMIP-1 and OMIP-2 (based on the eight models), respectively. Tsujino et al. (2020)

suggested that the lower salinity in upper Arctic Ocean in OMIP-2 relative to OMIP-1 is partly caused by the difference in salinity to which sea surface salinity is restored between OMIP-1 and OMIP-2. Sea ice decline can increase liquid freshwater accumulation in the Beaufort Gyre, by both supplying sea ice meltwater and increasing convergence of other freshwater

components (Wang et al., 2018b). Tsujino et al. (2020) shows that sea ice volume in OMIP-2 has a larger negative trend than OMIP-1, so the larger sea ice decline in OMIP-2 can also partly contribute to the higher freshwater content in OMIP-2. Positive freshwater content biases are also reported in CMIP5 and CMIP6 fully-coupled models (Shu et al., 2018; Zanowski et al., 2021; Wang et al., 2022b). OMIP models also have large inter-model spreads in the freshwater content simulations (Figs. S3 and S4). CMCC-CM2-SR5, CMCC-ESM2, and GFDL-OM4p5B in OMIP-1 and CMCC-CM2-SR5, CNRM-CM6-

1 and MRI-ESM2-0 in OMIP-2 have too much freshwater content in the Arctic Ocean, while IPSL-CM6A-LR and NorESM2-LM in OMIP-1 have too little freshwater content, which are consistent with their salinity biases (Fig. 5).

Despite the model spreads in the simulated mean state, OMIP-1 and OMIP-2 models show good agreement on the inter-annual and decadal variability of liquid freshwater content in the Arctic Basin (Fig. 8). OMIP simulations indicate a negative trend in freshwater content in the Arctic Basin from the 1960s to the mid-1990s, and then a positive trend afterwards which

is consistent with observations (Rabe et al., 2014; Wang et al., 2019b). Previous studies show that an unprecedented amount of freshwater has accumulated in the Amerasian Basin since the 1990s (Rabe et al., 2014; Polyakov et al., 2013; Wang et al., 2019b; Wang and Danilov, 2022; Solomon et al., 2021; Proshutinsky et al., 2019), and the Arctic total liquid freshwater content became relatively stable in the last decade (2010s) (Solomon et al., 2021). These decadal changes are reproduced by OMIP-2 simulations (Fig. 8). However, the overall positive trend since the 1990s in OMIP is relatively weaker than the

observations (Fig. 8b), which was also found in CORE-II simulations (Wang et al., 2016b).

### 3.3 Changes in Atlantic Water layer

Arctic Ocean Atlantic Water layer temperature varies on different time scales (Polyakov et al., 2004). Its fluctuations are linked to the highly variable nature of the Atlantic Water inflows, with abrupt cooling/warming events (Polyakov et al., 2020a). A warming pulse was observed in the 1990s in the Eurasian Basin and then in the Amerasian Basin in 2000s (Steele

and Boyd, 1998; Polyakov et al., 2012). It is about 1 ℃ warmer in the 1990s than in the 1970s in the Eurasian Basin based on observations (Polyakov et al., 2020a). Figures 9 and S5 show that OMIP-1 models are able to reproduce a similar warming event, but it is much weaker than the observations. In OMIP-2 simulations, the warming event in the Eurasian Basin is also much weaker than the observed (Figs. 9 and S6). The multi-model means produced by both the OMIP groups do not show decadal warming in the Amerasian Basin during the 1990s. (Fig. 9). The model spreads in simulating the

warming event in the 1990s are large, with some models being able to reproduce the decadal variability and some models not (Figs. S5 and 6).

Polyakov et al. (2020a) found that the Atlantic Water layer temperature increased rapidly in the 2000s, and then reached a temporary quasi-equilibrium afterwards. The rapid Atlantic Water layer warming in this period can be simulated by OMIP-2

models, which cover the last decade in the simulations. However, in both OMIP-1 and OMIP-2, the rapid Atlantic Water layer warming in the 2000s is not as prominent as found in observations (Figs. 9 and S5).

The agreement on the inter-annual and decadal variability of Atlantic Water layer temperature in OMIP-1 and OMIP-2 is lower than liquid freshwater content (Figs. 8 and 9). One reason is the drastic influence of the remarkable warming at the end of the preceding simulation cycle on the following cycle, which is more pronounced in OMIP-2 because the temperature is higher in 2018 than in 2009 (the years that were used to initialize new simulation cycles in OMIP-2 and OMIP-1, respectively). All the OMIP-2 models show warmer Arctic basins in the first two decades than in the succeeding decade (Fig S6), indicating that the re-initialization can have impacts for at least two decades. The impact of the re-initialization is even larger than the natural decadal variability (Fig. 9c). A warm episode developed in the 1960s in the Amerasian Basin in OMIP-2 because the high temperature in the Eurasian Basin inherited from the preceding simulation cycle propagated downstream into the Amerasian Basin (Fig. 9d).

**3.4 Stratification**

To evaluate the ocean stratification, we compared surface mixed layer depth and cold halocline base depth with the observations in Figs. 10 and 11. Mixed layer depth is important for Arctic physical, chemical and biological processes (Peralta-Ferriz and Woodgate, 2015). In this study, it is defined as the depth where the potential density is larger than the surface density by 0.1 kg/m$^3$, which is considered a suitable threshold criterion for calculating surface mixed layer depth in the Arctic Ocean (Peralta-Ferriz and Woodgate, 2015). Arctic Ocean mixed layer depth shows a remarkable seasonal cycle, deeper in winter and shallower in summer. Considering that the summer mixed layer depth in many areas of the Arctic Ocean is ~10 m based on observations (Peralta-Ferriz and Woodgate, 2015), which is quite close to the thickness of the first vertical layer in many OMIP models, we only study mixed layer depth in winter in this paper.

Using the observations during 1979–2012, Peralta-Ferriz and Woodgate (2015) quantified Arctic Ocean surface mixed layer depth for six Arctic regions (Southern Beaufort Sea, Canada Basin, Chukchi Sea, Makarov Basin, Eurasian Basin and Barents Sea), and their estimate shows that cold season (November–May) mixed layer depths in these six regions are 29.0, 33.1, 34.6, 52.0, 72.5, and 168 m, respectively. So mixed layer depth is deepest in the Barents Sea, followed by the Eurasian Basin and Makarov Basin, and is relatively shallow in the southern Beaufort Sea, the Canadian Basin and the Chukchi Sea. Figure 10 shows that both the OMIP-1 and OMIP-2 multi-model mean results can reproduce this spatial pattern. The discrepancy of mixed layer depth between simulations and observations in the Chukchi and Beaufort Seas is relatively small. This may be partially attributed to the good performance in the representation of Pacific Water volume transport through the Bering Strait in OMIP models (see the results in Section 3.6.1). However, the simulated mixed layer depth in the Eurasian Basin is shallower than the observations, especially in the OMIP-1 experiments. Figures S7 and S8 indicate that both OMIP-1 and OMIP-2 models have quite large inter-model spreads in the simulated mixed layer depth, which is similar to the situation of the CORE-II simulations (Ilicak et al., 2016).

Figure 10c shows that the cold season mixed layer depth has positive trends in most of the Arctic Ocean over the last 40 years in the OMIP-2 multi-model mean, except for the Norwegian Sea, Baffin Bay, southern Barents Sea, and part of the Greenland Sea where the trends are negative. The negative trends along the Atlantic Water pathway are mainly caused by less ocean surface heat release in a warming climate (Shu et al., 2021). Some models (CMCC-CM2-HR4, CMCC-CM2-SR5, CNRM-CM6-1, and EC-Earth3) simulate episodic deep convection (maximum of mixed layer depth deeper than 200 m) in the Nansen Basin (Fig. S9). It might bring oceanic heat from the Atlantic Water layer to the mixed layer and reduce sea ice thickness in these models.

The Arctic cold halocline layer is an important insulator between the warm Atlantic Water layer and the cold surface mixed layer and sea ice above. The cold halocline base depth used here is defined as the depth where the ratio of the density gradient due to temperature to the density gradient due to salinity equals 0.05 (Bourgain and Gascard, 2011), that is:

$$R_\rho = (\alpha\, \partial\theta/\partial z)/(\beta\, \partial S/\partial z) = 0.05, \tag{2}$$

where $\alpha$, $\beta$, $\theta$, and S are the thermal expansion coefficient, haline contraction coefficient, potential temperature and salinity, respectively. This depth characterizes the transition from halocline to thermocline (Bourgain and Gascard, 2011).

The Arctic cold halocline base depth derived from the PHC3.0 climatology and OMIP simulations are shown in Fig. 11. It is shallow in the Eurasian Basin and deep in the Amerasian Basin according to PHC3.0 (Fig. 11a), and this spatial pattern can be qualitatively reproduced by OMIP models (Fig. 11b and c). However, the simulated halocline base depth is too deep in both the Eurasian and Amerasian Basins. In most area of the Eurasian Basin, it is shallower than 100 m based on PHC3.0, but it is deeper than 120 m in the OMIP simulations. It is shallower than 210 m based on PHC3.0 in the Canada Basin, while it is deeper than 300 m in the OMIP simulations. The inter-model spreads are also quite large in both OMIP-1 and OMIP-2 models (Figs. S10 and S11). FIO-ESM-2-0 and AWI-CM-1-1-LR perform the best in OMIP-1 and OMIP-2, respectively. Surface mixed layer depth in AWI-CM-1-1-LR also fits the observations well (Fig. S8). So, AWI-CM-1-1-LR in OMIP-2 has a good overall performance in the stratification simulations. Observations indicate that the cold halocline layer in the Eurasian Basin has a thinning trend recently (Polyakov et al., 2020c, b). This trend can be reproduced by the OMIP-2 multi-model mean result (Fig. 11d) and each OMIP-2 individual model (not shown). The OMIP-2 multi-model mean result also shows the cold halocline base depth in the Amerasian Basin has a positive anomaly during 2009–2018 relative to its climatology (Fig. 11d), consistent with freshwater accumulation in the upper Amerasian Basin since mid-1990s (Fig. 8). The positive anomaly of the simulated cold halocline base depth in the Amerasian Basin agrees well with the observed positive trend since 1970 (Muilwijk et al., 2022), which is not surprising since the models largely reproduced the observed freshwater accumulation in the Arctic Basin (Fig. 8b).

**3.5 Sea surface height**

Changes in sea surface height in the Arctic Basin reflect the variation of liquid freshwater content (e.g., Morison et al., 2012; Wang et al., 2021). Furthermore, interannual changes in sea surface height are good indicators of interannual changes in the upper Arctic Ocean circulation (Morison et al., 2021), because surface geostrophic currents dominate the Arctic surface

velocity on spatial scales larger than 10 km and timescales longer than a few days (Doglioni et al., 2023). To evaluate the mean state and variability of upper ocean circulation, we compare modelled sea surface height with observational estimates from altimetry measurements provided by Armitage et al. (2016).

The Arctic sea surface height is featured with a high in the Beaufort Sea associated with the anticyclonic Beaufort Gyre, a low in the Greenland Sea associated with the cyclonic Greenland Sea gyre, and a large-scale gradient associated with the Transpolar Drift stream (Fig. 12a) (Armitage et al., 2016). The multi-model mean results of both OMIP-1 and OMIP-2 can reproduce these main features of the sea surface height in the Arctic (Fig. 12b and 12c). However, OMIP simulations have a broader and weaker Beaufort Gyre than the observed, and its center is biased toward the Eurasian Basin. The Beaufort Gyre in OMIP-1 is weaker than that in OMIP-2 (Fig. 12), which is consistent with lower freshwater column in the Beaufort Gyre in OMIP-1 than in OMIP-2 (Fig. 7). The strength of the Beaufort Gyre and the location of its center have large inter-model spreads in OMIP models (Figs. S12 and S13). The multi-model mean cyclonic Greenland Sea gyre in both OMIP-1 and OMIP-2 is also weaker than the satellite observation (Fig. 12), with large inter-model spreads as well (Figs. S12 and S13).

Major changes have occurred in the upper ocean circulation in the Arctic during the first two decades of the 21st century (Wang and Danilov, 2022), mainly manifested by the unprecedented spin-up of the Beaufort Gyre. Satellite-derived sea surface height shows a marked spin-up of the Beaufort Gyre in the period of 2004–2009 (Fig. 13a), which was associated with the anomalous negative wind curl over the Canada Basin in this period. Satellite observations also show a reduction in the sea surface height from 2004 to 2009 in the Makarov and Eurasian basins, and an increase in the Laptev and East Siberian seas. In the period of 2009 to 2014, both the Beaufort High and Arctic Oscillation were close to neutral states on average, and a positive sea level pressure anomaly was centered over the outer shelf of the East Siberian Sea (Wang and Danilov, 2022). Consistently, satellite observations show a reduction in the Beaufort, Laptev and East Siberian Seas, and an increase in the Makarov Basin and over the outer shelf of the East Siberian Sea in this period (Fig. 13b). For the period of 2014–2018, a negative wind curl anomaly in the southern part of the Canada Basin caused a spin-up of the Beaufort Gyre, which was confined to the southern Canada Basin in this period (Wang and Danilov, 2022; Fig. 13a and 13c). These changes in the sea surface height can be largely reproduced by the multi-model mean results of both OMIP-1 and OMIP-2 (Fig. 13d–g). However, the magnitude of the changes is underestimated by the multi-model mean, and the simulated changes in the Beaufort Gyre are also biased toward northwest. The observed sea surface height reduction in the Beaufort Sea from 2009 to 2014 is not reproduced by the OMIP-2 multi-model mean (Fig. 13f).

### 3.6 Gateway transports

In this subsection, the simulated ocean volume transport, heat transport, and liquid freshwater transport through the Bering Strait, BSO, Fram Strait, and Davis Strait are evaluated. Ocean volume transport OVT, heat transport OHT, and liquid freshwater transport FWT through each Arctic Ocean gateway are calculated as follows:

$$OVT = \int_{-H(\lambda)}^{0} \int_{\lambda_1(z)}^{\lambda_2(z)} v \, d\lambda \, dz, \tag{3}$$

$$OHT = \rho_o c_p \int_{-H(\lambda)}^{0} \int_{\lambda_1(z)}^{\lambda_2(z)} v(\theta - \theta_{ref}) d\lambda dz, \tag{4}$$

$$FWT = \int_{-H(\lambda)}^{0} \int_{\lambda_1(z)}^{\lambda_2(z)} v(1 - S/S_{ref}) d\lambda dz, \tag{5}$$

where $v$ is ocean velocity normal to the section of each gateway, $\theta$ is the potential temperature, $\rho_o$ is seawater density, $c_p$ is the specific heat capacity of seawater, $\theta_{ref}$ is the reference temperature set to be 0 ℃, $S$ is the salinity, $S_{ref}$ is the reference salinity set to be 34.8 psu, $H$ is water depth, and $\lambda$ is the distance along the gateway transect.

### 3.6.1 Ocean volume transport

The mean net volume transport through the Bering Strait based on observations is 0.8±0.2 Sv (1 Sv ≡ $1\times10^6$ m$^3$/s) during 1990 to 2007 (Roach et al., 1995; Woodgate and Aagaard, 2005) and 1.0±0.05 Sv during 2003 to 2015 (Woodgate, 2018a). The multi-model mean results of OMIP-1 and OMIP-2 in their last cycles are 1.0±0.1 and 1.1±0.1 Sv (Tables 2 and 3), respectively. So, the climatology volume transport through the Bering Strait is reasonably simulated by both OMIP-1 and OMIP-2. However, the observed positive trend in recent decades is not correctly reproduced in OMIP simulations. Observations indicate that the volume transport has a positive trend (0.01 Sv/year) during 1990 to 2019 (Woodgate and Peralta-Ferriz, 2021), while the trends in all OMIP simulations are negative since 1990 (Figs. 14a, S14 and S15). There is no evidence that the simulated erroneous trends are related to model horizontal or vertical resolutions. Models not part of OMIP and employing different atmospheric forcing have the same issue (Nguyen et al., 2020). The observed upward volume transport trend is also not reproduced in historical simulations of CMIP6 fully-coupled models (Zanowski et al., 2021, Wang et al., 2022b). The OMIP models simulated a sea surface height drop throughout the northern Bering Sea in 2009–2014 relative to 2003–2008, with the largest decrease in the eastern Bering Sea (Fig. 15b). Satellite observations, on the contrary, revealed a sea surface height increase in most of the northern Bering Sea except for the Norton Sound for the same period (Fig. 15a). Furthermore, the sea surface height increase in the western Chukchi Sea is larger in the models than in the observation. The sea surface height gradient between the eastern Bering Sea and western Chukchi Sea is one important factor controlling the variability of the Bering Strait throughflow (Danielson et al., 2014; Peralta‐Ferriz and Woodgate, 2017; Zhang et al., 2020). Therefore, the reason for the discrepancy between the simulated and observed throughflow trends may be caused by the unrealistic reduction of sea surface height in the Bering Sea and the overestimated increase in the western Chukchi Sea as well in the 2010s in the OMIP-2 simulations. It remains to explore the reasons for the unrealistic representation of sea surface height changes in these regions in the future.

The mean net volume transport through the BSO is ~2.0 Sv based on historical observations during 1997 to 2007 (Smedsrud et al., 2010, 2013). The multi-model mean results of OMIP-1 and OMIP-2 in their last cycles are 3.0±0.5 and 3.3±0.4 Sv (Tables 2 and 3), respectively. So this transport is overestimated in both OMIP-1 and OMIP-2 experiments (Fig. 14b). Eight models (CanESM5, CanESM5-CanOE, CMCC-CM2-HR4, CMCC-CM2-SR5, CMCC-ESM2, CNRM-CM6-1, EC-Earth3, and IPSL-CM6A-LR) employing the NEMO ocean model produce relatively large net volume transport through the BSO compared with other models. As a result, the ensemble mean may be biased toward NEMO-family models. The trends of the

BSO net volume transport in OMIP simulations over the simulated periods are not significant (Fig. 14b). There is no observation-based estimation about the net volume trend, but Skagseth et al. (2020) estimated the volume flux of Atlantic Water inflow to the Barents Sea based on an array of current meters in the western Barents Sea. They found that it increased by 0.2 Sv over the period of 1998 to 2018 although the trend is not statistically significant.

The mean net volume transport through the Fram Strait is −2.0±2.7 Sv based on observations during 1997 to 2006 (Schauer et al., 2008). The large uncertainty in the observations may be caused by the fact that the volume transports of both inflow and outflow through the Fram Strait are relatively large. The multi-model mean results of OMIP-1 and OMIP-2 are −2.3±0.5 and −2.6±0.4 Sv (Tables 2 and 3), respectively. They are within the range of observational uncertainty.

The mean net volume transport through the Davis Strait is −1.6±0.5 Sv during 2004 to 2010 (Curry et al., 2014). The multi-model mean results of OMIP-1 and OMIP-2 in their last cycles are −1.6±0.3 and −1.8±0.4 Sv (Tables 2 and 3), respectively. So the multi-model mean results fit the observations well in both OMIP-1 and OMIP-2. OMIP-2 simulated a decadal increase in ocean volume export in the Davis Strait in the 2010s (Fig. 14d), which was induced by the dynamic sea level drop south of Greenland in this period (Wang et al., 2022a).

Overall, the climatology of the net volume transports through the Arctic Ocean gateways is well represented in the multi-model mean results of OMIP simulations. However, Tables 2 and 3 indicate that there are large inter-model spreads in OMIP models. Figures S14 and S15 indicate that OMIP models are consistent in the inter-annual and decadal variability, and the variability in these two versions of the OMIP simulations also agrees well.

### 3.6.2 Ocean heat transport

Ocean heat transport through the Bering Strait is ~9.5–19.0 TW computed using $\theta_{ref} = -1.9\,°C$ based on observations during 2001 to 2015 (Woodgate, 2018a). The multi-model mean results of OMIP-1 and OMIP-2 in their last cycles are 3.5±1.7 and 3.0±1.9 TW computed using $\theta_{ref} = 0\,°C$ (Tables 4 and 5), and 11.4±2.1 and 11.9±2.3 TW computed using $\theta_{ref} = -1.9\,°C$, respectively. Thus the simulated mean ocean heat transport in OMIP fits the observations. A positive trend (0.35±0.17 TW/year) during 2000 to 2018 has been found based on observations (Woodgate and Peralta-Ferriz, 2021). The OMIP models also obtained positive trends, but much weaker (0.13±0.26 TW/year in OMIP-2) (Fig. 14e). The weak trend is possibly caused by the erroneous negative trend in ocean volume transport in OMIP simulations shown in Fig. 14a.

Ocean heat transport through the BSO is the largest among the four gateways. Net ocean heat transport through the BSO is 48 and 74 TW based on the estimations by Skagseth et al. (2008) and Smedsrud et al. (2010), respectively. The multi-model mean results of OMIP-1 and OMIP-2 in their last cycles are 66.2±11.6 and 73.9±11.6 TW (Tables 4 and 5), respectively. Consistent with the volume transport, the eight NEMO-family models simulate relatively large ocean heat transport through the BSO compared with other models. Similar behaviours are found in the CMIP6 fully-coupled climate models with NEMO as their ocean components in both historical simulations and future projections (Pan et al., 2023). It might be related to the different vertical mixing scheme (TKE turbulent closure scheme) or the representation of bathymetry in NEMO-family models. Overall, OMIP multi-model mean results fit the observations for the BSO. Its positive trend since 1980 reported by

Skagseth et al. (2008), Årthun et al. (2012) and Wang et al. (2019a) can also be reproduced by both OMIP-1 and OMIP-2 (Fig. 14f).

Based on the observational estimation by Schauer et al. (2008), ocean heat transport in the Atlantic Water (warmer than 1 ℃) through the Fram Strait is between 26 and 50 TW without significant trend during 1997 to 2006. The multi-model mean net ocean heat transports of OMIP-1 and OMIP-2 in the last cycle are $21.2 \pm 3.7$ and $20.6 \pm 6.0$ TW (Tables 4 and 5), respectively. They are at the lower end of the observation-based estimate, which was also founded in CORE-II simulations (Ilicak et al., 2016). One possible reason may be that the observation-based estimate mentioned above is the heat transport in the Atlantic Water inflow but not the net ocean heat transport through the whole gateway. Most CMIP6 fully-coupled models also exhibit similarly low net ocean heat transport through the Fram Strait (Heuzé et al., 2023). There is a clear increase in the heat transport in the 2010s in OMIP-2 (Fig. 14g), consistent with previous studies (Wang et al., 2020).

Ocean heat transport through the Davis Strait is $18 \pm 17$ and $20 \pm 9$ TW based on observations from 1987 to 1990 and from 2004 to 2005 (Cuny et al., 2005; Curry et al., 2011), respectively. The multi-model mean results of OMIP-1 and OMIP-2 in their last cycles, being $12.6 \pm 2.4$ and $13.8 \pm 3.1$ TW (Tables 4 and 5), respectively, underestimated the observed values.

Tables 4 and 5 show that OMIP models have large inter-model spreads in the simulations of the mean ocean heat fluxes at all the four Arctic Ocean gateways. For the inter-annual and decadal variability, OMIP-1 and OMIP-2 models have relatively good agreement between their multi-model means (Fig. 14), although a few individual models show variability distinct from others for the Davis and Fram straits (Figs. S14 and S15).

### 3.6.3 Freshwater transport

Ocean freshwater transport through the Bering Strait is $(2.4 \pm 0.3) \times 10^3$ km$^3$/year from 1990 to 2004 and $(2.3–3.5) \times 10^3$ km$^3$/year between 2001 and 2015 based on observations (Woodgate and Aagaard, 2005; Woodgate, 2018a). Woodgate and Peralta-Ferriz (2021) also found that it has a significant positive trend [$35 \pm 17$ (km$^3$/year)/year] during 1990 to 2019 due to both the increase in ocean volume transport and the decrease in seawater salinity. The multi-model mean results of OMIP-1 and OMIP-2 in the last cycle are $(2.1 \pm 0.3) \times 10^3$ and $(2.4 \pm 0.3) \times 10^3$ (km$^3$/year) (Tables 6 and 7), respectively. So OMIP models can reproduce the mean freshwater transport before 2004, but after that they significantly underestimate the freshwater transport and show incorrect negative trends (Figs. 14i and S15i), mainly due to the negative trend in the simulated ocean volume transport (Figs. 14a and S15a).

Freshwater transport through the BSO is a freshwater sink for the Arctic Ocean with the annual mean value of $-0.09 \times 10^3$ km$^3$/year based on historical observations (Serreze et al., 2006; Haine et al., 2015). The simulated freshwater transports are $(-0.58 \pm 0.22) \times 10^3$ and $(-0.48 \pm 0.18) \times 10^3$ km$^3$/year in OMIP-1 and OMIP-2 (Tables 6 and 7), respectively. The biases of the freshwater transport in the OMIP simulations may be caused by the overestimation of ocean volume transport (Fig. 14b).

Liquid freshwater transport through the Fram Strait is $(-2.70 \pm 0.53) \times 10^3$ km$^3$/year based on historical observations (Serreze et al., 2006; Haine et al., 2015). It is also a freshwater sink of the Arctic Ocean. The multi-model mean results of OMIP-1 and OMIP-2 underestimated the observation, being $(-1.72 \pm 0.35) \times 10^3$ and $(-2.16 \pm 0.52) \times 10^3$ km$^3$/year (Tables 6 and 7),

respectively. A strong increase in freshwater export at the beginning of the 2010s was simulated in OMIP-2 (Fig. 14k),

consistent with the observed changes (de Steur et al., 2018).

Ocean freshwater transport through the Davis Strait is $(-2.93\pm0.19)\times10^3$ km$^3$/year based on historical observations during 2004 to 2010 (Curry et al., 2014). It is also a freshwater sink of the Arctic Ocean. The multi-model mean results of OMIP-1 and OMIP-2 in their last cycles are $(-2.54\pm0.46)\times10^3$ and $(-3.27\pm0.62)\times10^3$ km$^3$/year (Tables 6 and 7), respectively. Tables 6 and 7 also indicate that the eight NEMO-family models simulate relatively large freshwater export through the Davis Strait

compared with other models. The freshwater export in the Davis Strait increased in the 2010s in OMIP-2, owing to the increase in ocean volume export (Fig. 14d and l).

The freshwater transports in the multi-model means correlate well between OMIP-1 and OMIP-2 in their common simulation period. Similar to ocean volume and heat transports, ocean freshwater transport also has large inter-model spreads in OMIP simulations (Tables 6 and 7), but most OMIP models share similar inter-annual and decadal variability (Figs. S14

and S15). The freshwater flux through the Bering Strait has the best agreement among the models (Figs. S14 and S15), but its recent trend is incorrectly simulated in all the OMIP models (Figs. S14i and S15i).

## 4 Discussion and Conclusions

In this work we assessed the Arctic Ocean simulations in 19 global ocean-sea ice models participating in CMIP6 OMIP (OMIP-1 and/or OMIP-2) (Griffies et al., 2016). The models used the same specified atmospheric forcing data sets and bulk

formula for surface flux calculations following the OMIP protocol. CORE2 forcing during 1948 to 2009 (Large and Yeager, 2009) and JRA55-do forcing during 1958 to 2018 (Tsujino et al., 2018) are the atmospheric forcing for OMIP-1 and OMIP-2 simulations, respectively. Modelled results of mean hydrography, liquid freshwater content, changes in Atlantic Water layer, upper ocean stratification, sea surface height, and gateway transports from the last cycle of the simulations were compared to the available observations and between the two OMIP versions.

Based on our evaluation, we concluded the following:

(1) For the simulations of the Arctic Ocean mean hydrography, most (but not all) models in OMIP can reproduce the vertical structures of temperature in the Arctic Ocean but with large biases and inter-model spreads, especially in the Atlantic Water layer depth range. The signal of warm Atlantic Water disappears too rapidly along the advection pathway in both OMIP-1 and OMIP-2 simulations compared to the observations. Most OMIP models suffer from too thick and deep Atlantic Water

layer and fresh biases in the halocline (50–400 m). OMIP model performances are similar to CORE-II models in the representation of mean hydrography (Ilicak et al., 2016). The biases and inter-model spreads in OMIP are relatively small compared with those of CMIP6 fully-coupled models reported by Khosravi et al. (2022), which also indicates that part of biases of the Arctic Ocean simulations in the fully-coupled models is caused by the biases in the atmospheric component models (Hinrichs et al., 2021).

(2) For the simulations of the Arctic Ocean liquid freshwater content, OMIP-1 and OMIP-2 models show relatively good performance in simulating the inter-annual and decadal variability, being largely consistent with observations. However, they have large inter-model spreads in the simulation of the mean state. OMIP-2 has more liquid freshwater content than OMIP-1. The overall performance of OMIP-1 and OMIP-2 in the simulation of freshwater content is also similar to CORE-II (Wang et al., 2016b).

(3) For temperature changes in the Atlantic Water layer, there is some disagreement between simulations and observations and between OMIP-1 and OMIP-2. The warming of the Atlantic Water layer in the 1990s and especially in the 2000s in OMIP simulations is too weak compared to observations. The warming anomaly in the 2010s is captured in the OMIP-2 simulations. In addition, there are also clear effects of high temperature at the end of the preceding simulation cycles on the following cycle, which are more pronounced in OMIP-2.

(4) For the simulations of the Arctic Ocean stratification, the climatology of cold season surface mixed layer depth can be well reproduced by the multi-model mean results of both OMIP-1 and OMIP-2, while the multi-model mean cold halocline base depths simulated by both OMIP-1 and OMIP-2 are much deeper than the PHC3.0 climatology, consistent with their fresh biases in the halocline. Fully-coupled climate models have similar fresh biases in the halocline (Wang et al., 2022b), so the potential impacts of future shoaling of the halocline base in a warming climate (Shu et al., 2022) on the atmosphere-
ocean-sea ice interactions may be biased low in climate model projections. OMIP models also have large inter-model spreads in the simulation of the Arctic Ocean stratification, which is similar to the performance of the CMIP6 fully-coupled models (Muilwijk et al., 2022). OMIP-2 models reasonably reproduced the observed shoaling of the halocline base in the eastern Eurasian Basin and the deepening in the Amerasian Basin over the last decade.

(5) The OMIP models can largely reproduce the satellite-observed changes in the sea surface height in the Arctic Basin,
implying that changes in the circulation of the upper Arctic Ocean are captured in the models. However, the magnitudes of the changes in the sea surface height are underestimated.

(6) For the simulations of Arctic Ocean gateway transports, the climatology of the net volume transports through Arctic Ocean gateways is well reproduced by the multi-model mean results of the OMIP simulations, although with large inter-model spreads. Relatively large biases are found in the climatological mean states of heat transport through the Fram Strait
and freshwater transport through the BSO and Fram Strait. OMIP models are relatively good at representing the inter-annual and decadal variability of volume, heat and freshwater transports through the four Arctic Ocean gateways. They have the best agreement in the simulated variability at the Bering Strait, but the recent upward trends in the Bering Strait fluxes are not captured by both OMIP-1 and OMIP-2. Considering the important implications of Bering Strait inflow for the Arctic heat, freshwater and nutrients, this issue should be fixed to obtain more realistic Arctic Ocean simulations in global ocean-
sea ice models and fully-coupled models.

(7) The OMIP models can better represent the interannual and decadal variability of Arctic Ocean gateway fluxes, freshwater content, sea surface height and upper ocean stratification than their mean states, similar to previous CORE-II models (Wang et al., 2016b).

Overall, no significant improvements are found in the Arctic Ocean simulations in global ocean-sea ice models stepping from CORE-II to OMIP. The previously found large biases and inter-model spread in the Atlantic Water layer simulations remain in OMIP-1 and OMIP-2. Therefore, it is not surprising that large biases and inter-model spread are found in the Arctic Ocean temperature and salinity simulations in CMIP6 fully-coupled models, and no significant improvements are found in the Arctic Ocean simulations from CMIP5 to CMIP6 (Khosravi et al., 2022; Wang et al., 2022b; Zanowski et al., 2021; Muilwijk et al., 2022; Heuzé et al., 2023). Improving model parameterizations (e.g. horizontal and vertical mixing) and using higher model resolutions may be possible solutions. However, to our knowledge, targeted studies on improving parameterizations for the Arctic Ocean simulations have been very limited, if any at all since the CORE-II project, and the horizontal resolutions in most OMIP models are still coarse (nominal 1 °, 24–50 km in the Arctic; Table 1). These factors may explain the lack of improvements from CORE-II to OMIP.

These 19 ocean-sea ice models employ various horizontal and vertical resolutions and vertical coordinates (Table 1), but we did not find obvious grouping of models in terms of horizontal/vertical resolutions or vertical coordinates for model skills. This may be partly caused by the fact that the majority of the OMIP models evaluated here have very coarse resolutions (nominal 1 °). Furthermore, previous studies also found that greatly enhanced horizontal resolution does not deliver unambiguous bias improvement in all regions for all models (Chassignet et al., 2020). We suggest that a dedicated high resolution Arctic Ocean model intercomparison project is needed.

According to the previous study (Shu et al., 2022), the Arctic Ocean warms faster than the global ocean mean in a warming climate, which is mainly contributed by the warming of the Atlantic Water layer. Considering the importance of the Atlantic Water in the Arctic Ocean climate change and the unrealistically deep and thick Atlantic Water layer simulated by ocean-sea ice and fully-coupled models, a high priority should be given to reduce the biases of Atlantic Water simulations in future model development. Wang et al. (2018) show that a model with 4.5-km (marginally eddy-permitting) resolution in the Arctic Ocean performs much better than a 24-km resolution model, especially in the simulations of the Atlantic Water layer, indicating that higher resolution (eddy permitting to eddy resolving) can help reduce model biases in Atlantic Water simulations through resolving eddy activity and reducing numerical mixing.

We did not find significant improvement in simulating Arctic Ocean using JRA55-do forcing than using CORE2 forcing. However, the simulated variability and trends of freshwater content and gateways transports agree well between OMIP-1 and OMIP-2. The CORE2 forcing dataset only contains atmosphere forcing and runoff till 2009 without being further updated. Therefore, JRA55-do forcing which has been updated to date (till 2022) is a good alternative to CORE2 forcing for studying recent changes in the Arctic Ocean in a model intercomparison framework until the modelling community agrees on newer forcing dataset for future OMIP. Part of the difference in the simulated temperature, salinity, and cold halocline base depth between OMIP-1 and OMIP-2 is caused by the design of the OMIP simulations. OMIP models run for no less than five cycles of the forcing periods as model spinup. Upon reaching the end of the year 2009 in OMIP-1 and 2018 in OMIP-2, the forcing is returned to 1948 in OMIP-1 and 1958 in OMIP-2. As the ocean climate state near the end of the forcing period is very different from the climatology of the 20$^{th}$ century, such as in OMIP-2 (jumping from 2018 to 1958), repeating the full

cycle of the atmosphere forcing can leave a large amount of Arctic Ocean heat and freshwater from the preceding simulation cycle to the following cycle. Our analysis suggests only repeating the atmosphere forcing of the 20th century in the model spinup cycles, which has relatively weak climate change signals.

**Code availability**

Matlab code used to process data and generate figures is available at https://zenodo.org/record/7808054#.ZC_h8fnv26s.

**Data availability**

OMIP-1 and OMIP-2 datasets are available at https://esgf-node.llnl.gov/search/cmip6/. The PHC3.0 is from http://psc.apl.washington.edu/nonwp_projects/PHC/Data3.html. Sea surface height from altimetry measurements is available at http://www.cpom.ucl.ac.uk/dynamic_topography.

**Author contribution**

QS, QW and CG proposed and led this evaluation study. QS, SW and YH processed the model outputs and produced the figures. All authors contributed to the writing and editing processes.

**Competing interests**

The authors declare that they have no conflict of interest.

**Acknowledgments**

Qi Shu was supported by the Chinese Natural Science Foundation (41941012 and 42276253), Shandong Provincial Natural Science Foundation (ZR2022JQ17), and the Taishan Scholars Program (No. tsqn202211264). Qiang Wang was supported by the Helmholtz Climate Initiative REKLIM (Regional Climate Change and Human) and the EPICA project in the research theme "MARE:N – Polarforschung/MOSAiC" funded by the German Federal Ministry for Education and Research with funding number 03F0889A. Chuncheng Guo acknowledges support from the Research Council of Norway funded projects KeyPOCP (328941) and KeyClim (295046). This work is a contribution to the UN Decade of Ocean Science for Sustainable Development (2021-2030) through both the Decade Collaborative Centre on Ocean-Climate nexus and Coordination amongst decade implementing partners in P. R. China (DCC-OCC) and the approved Programme of the Ocean to climate Seamless Forecasting system (OSF). OMIP-1 and OMIP-2 data are from https://esgf-node.llnl.gov/search/cmip6/. The PHC3.0 climatology dataset is downloaded from http://psc.apl.washington.edu/nonwp_projects/PHC/Climatology.html.

Satellite derived Arctic sea surface height is from http://www.cpom.ucl.ac.uk/dynamic_topography/. We would like to thank the above data providers and the two reviewers (Hannah Zanowski and Jonathan W. Rheinlænder) for their helpful
comments.

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

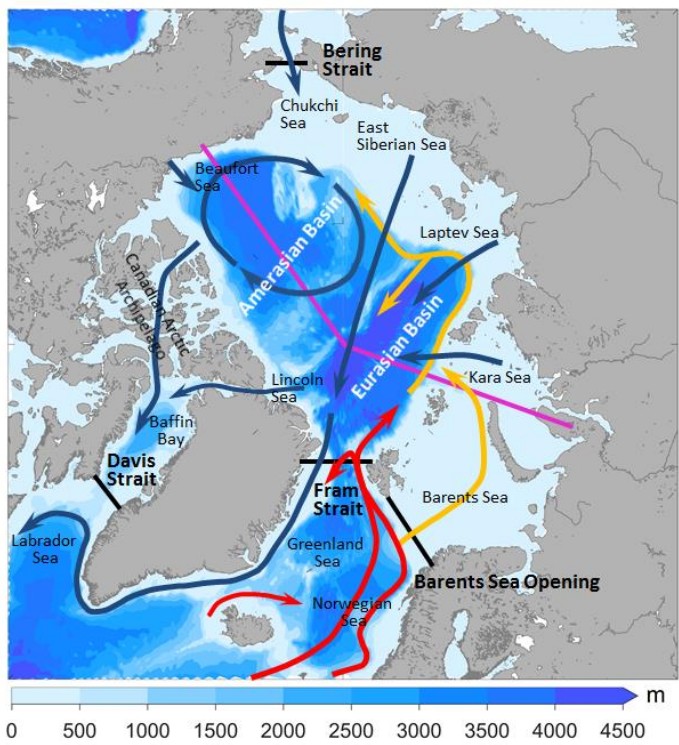

Figure 1. Schematic of main ocean circulations in the pan-Arctic Ocean. The freshwater circulation is shown with dark blue arrows, and the Atlantic Water circulation is shown with red/orange arrows. The black lines indicate the four Arctic gateways used in this study. The pink line along 70 °E and 145 °W crossing the North Pole indicates the location of section S used in Figs. 4, S1, and S2.

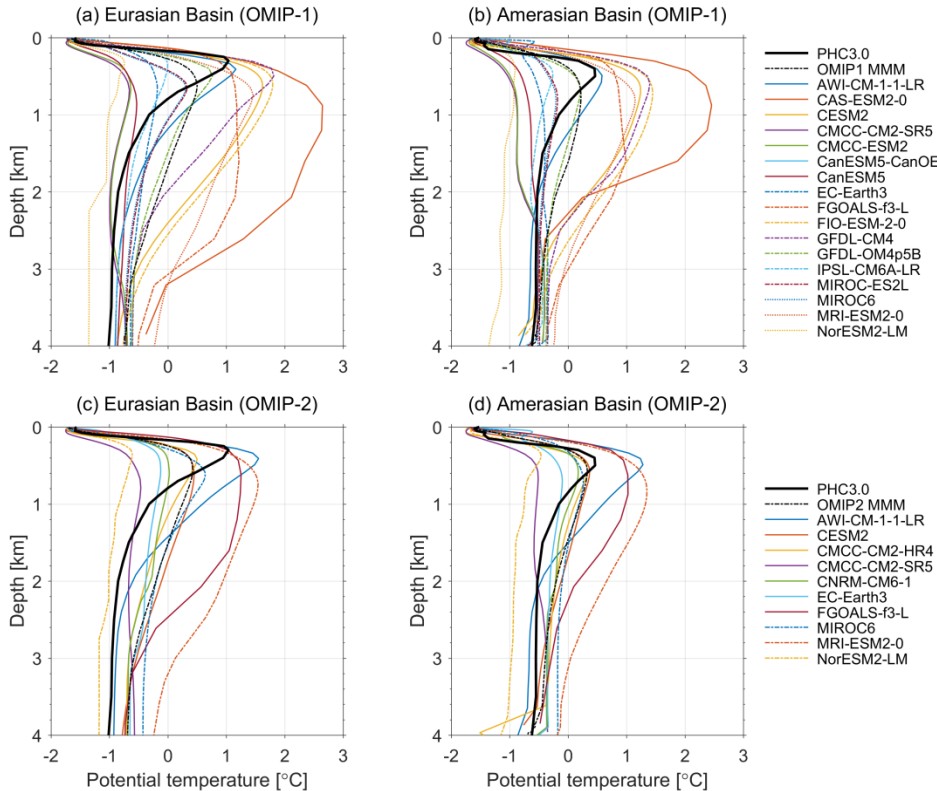

**Figure 2.** Potential temperature (unit: ℃) profiles averaged from 1971 to 2000 in the Eurasian (left) and Amerasian (right) Basins. MMM indicates multi-model mean.

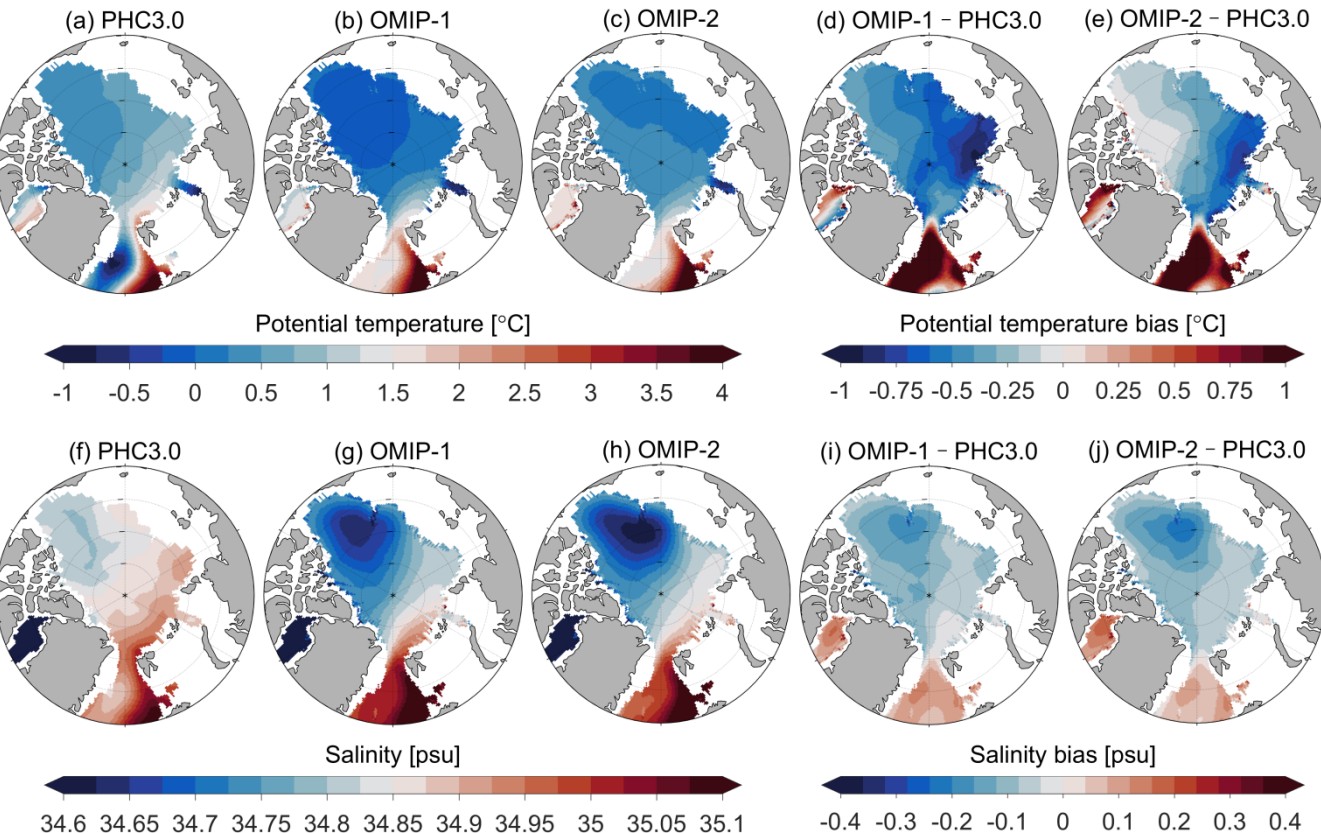

**Figure 3. (upper) Potential temperature and (bottom) salinity at 400 m from (a,f) PHC3.0, (b,g) OMIP-1, and (c,h) OMIP-2 multi-model mean results, and the biases of (d,e) potential temperature and (i,j) salinity of (d,i) OMIP-1 and (e,j) OMIP-2. The average over 1971–2000 is shown for OMIP-1 and OMIP-2. The eight models providing both OMIP-1 and OMIP-2 simulations (indicated with bold model ID in Table 1) are used here.**


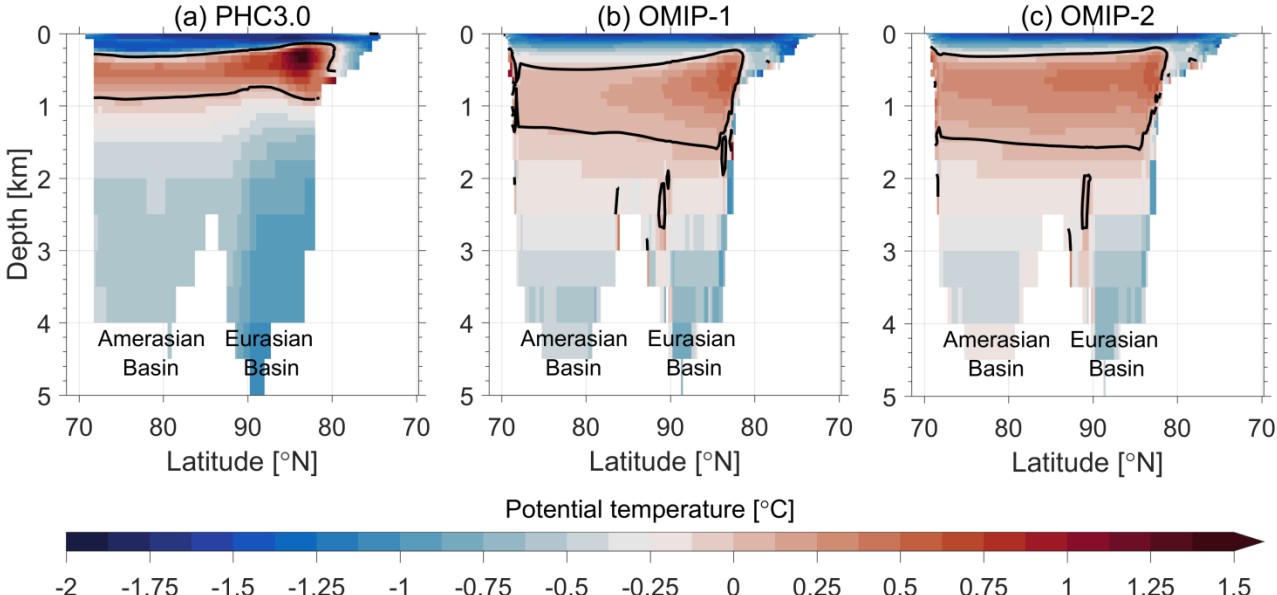

**Figure 4. Vertical section of potential temperature (unit: ℃) along the 70 ̊E–145 ̊W section S (indicated in Fig. 1) from (a) the PHC3.0 dataset, (b) OMIP-1, and (c) OMIP-2 multi-model mean results. The averages over 1971–2000 are shown for the model results. Black line is the 0 ℃ isotherm, which can be considered as the boundary of the Atlantic Water layer. The eight models providing both OMIP-1 and OMIP-2 simulations (indicated with bold model ID in Table 1) are used here.**


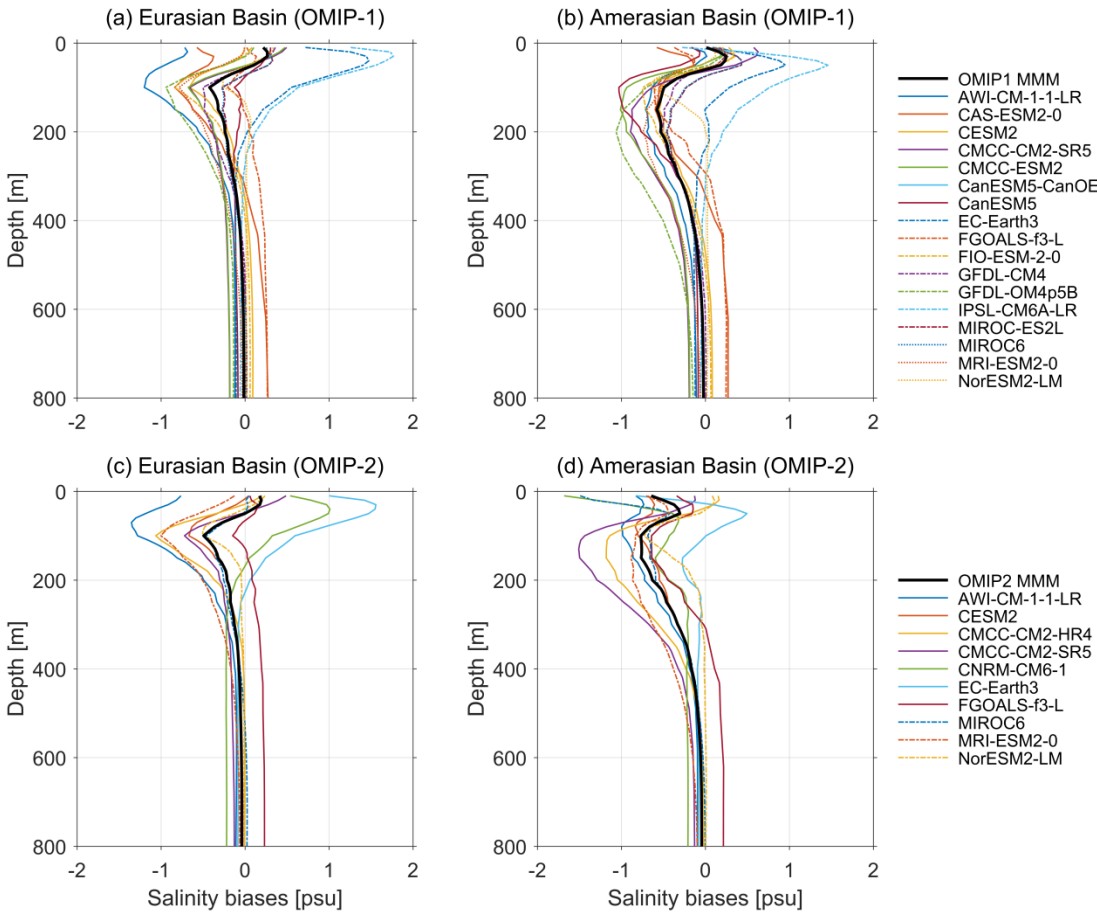

**Figure 5. The basin-mean biases of salinity (unit: psu) in the (left) Eurasian and (right) Amerasian basins. The biases are calculated as the difference between the 1971–2000 model mean and the PHC3.0. MMM indicates multi-model mean.**


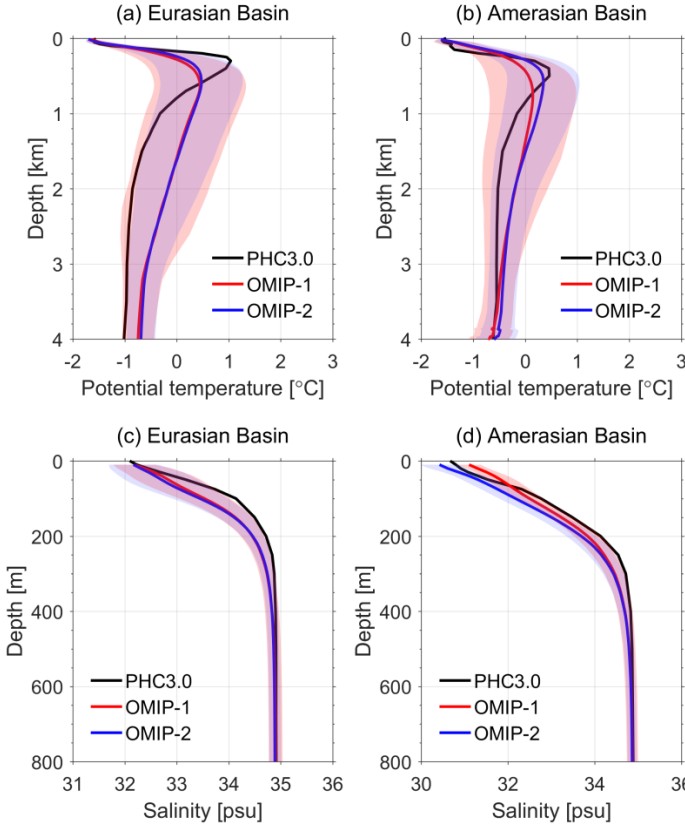

**Figure 6. Multi-model mean (MMM) basin-mean (upper) potential temperature (unit: ℃) and (lower) salinity (unit: psu) averaged from 1971 to 2000 in the (left) Eurasian and (right) Amerasian basins from OMIP-1 and OMIP-2 models. The model spreads (one standard deviation) are shown with shading areas. The eight models with bold model ID in Table 1 are used here.**

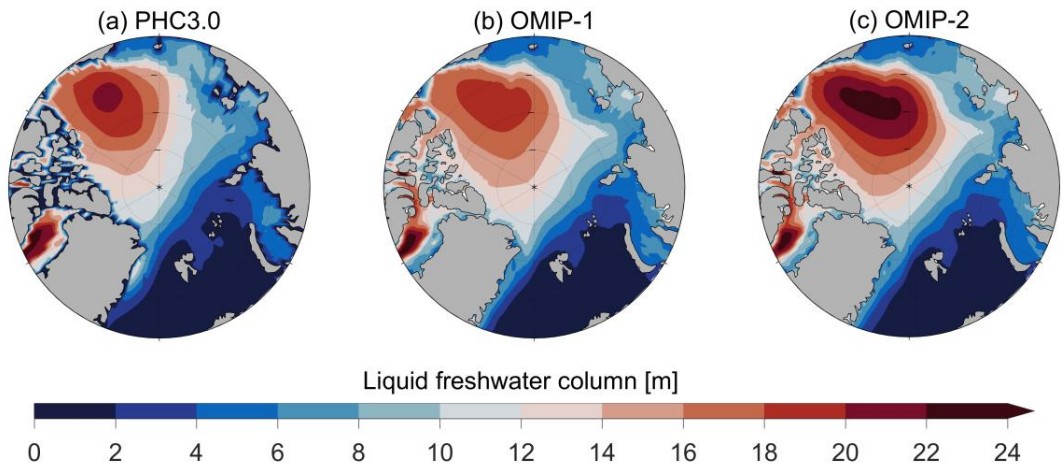


**Figure 7. Liquid freshwater column (m) in (a) PHC3.0, and (b) OMIP-1 and (c) OMIP-2 multi-model mean results. The model results are averaged over 1971–2000. The eight models with bold model ID in Table 1 are used here.**

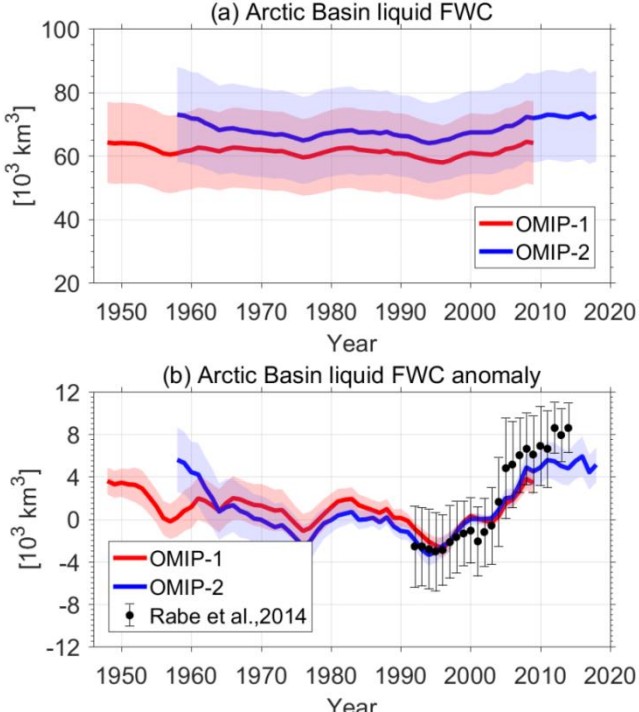

**Figure 8. Liquid freshwater content (FWC) (a) and its anomaly (b) in the Arctic Basin in OMIP-1 (red) and OMIP-2 (blue). The lines are the multi-model mean results, and the shading areas represent one standard deviation of the OMIP models. The eight models with bold model ID in Table 1 are used here. Observations are from Rabe et al. (2014) and Wang et al. (2019b).**

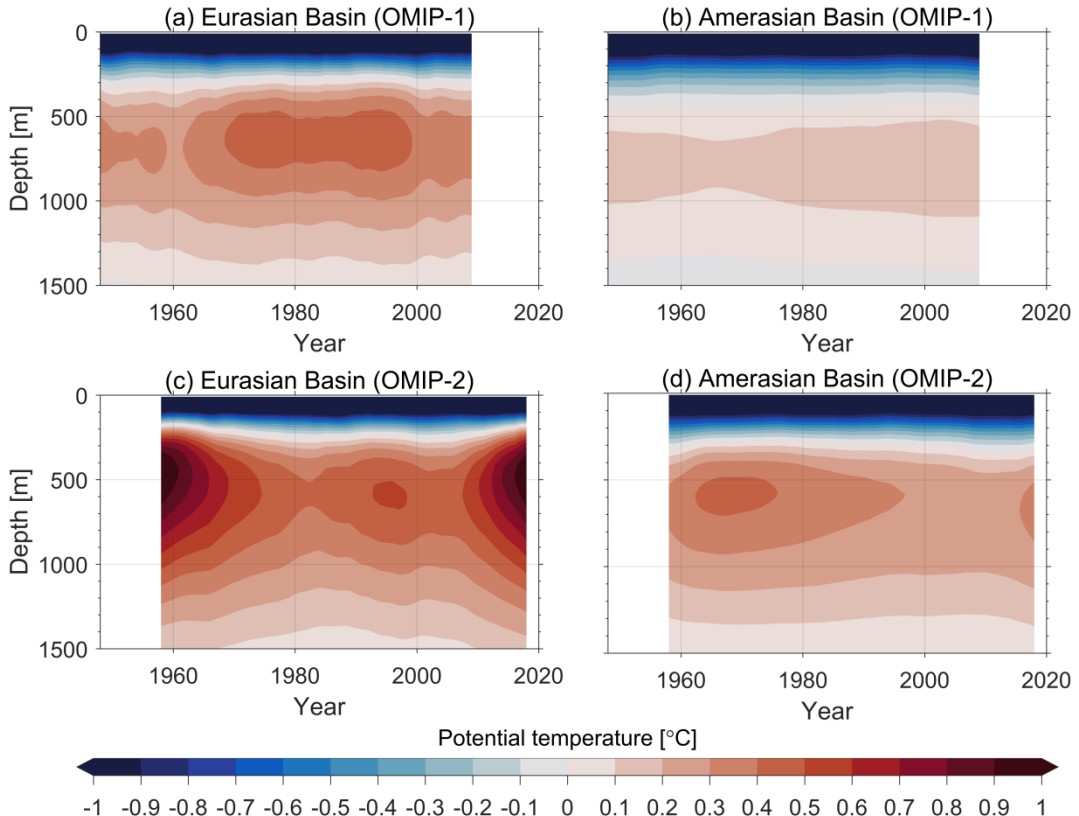

**Figure 9.** Hovmöller diagram of basin-mean potential temperature (unit: ℃) for the Eurasian Basin (a,c) and Amerasian Basin (b,d) in OMIP-1 (a,b) and OMIP-2 (c,d). The multi-model mean of the eight models with bold model ID in Table 1 are used here.

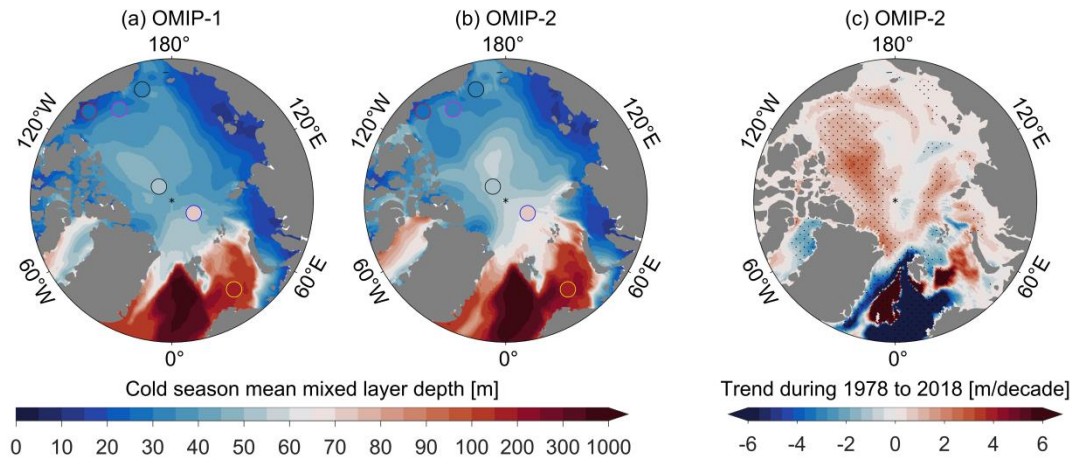

**Figure 10.** Cold season (November–May) mean mixed layer depth during 1979 to 2009 based on multi-model mean results: (a) OMIP-1, (b) OMIP-2. (c) Linear trend of cold season mixed layer depth during 1978 to 2018 in OMIP-2. The dots in (a) and (b) are observations during 1979 to 2012 (Peralta-Ferriz and Woodgate, 2015). The eight models with bold model ID in Table 1 are used here.

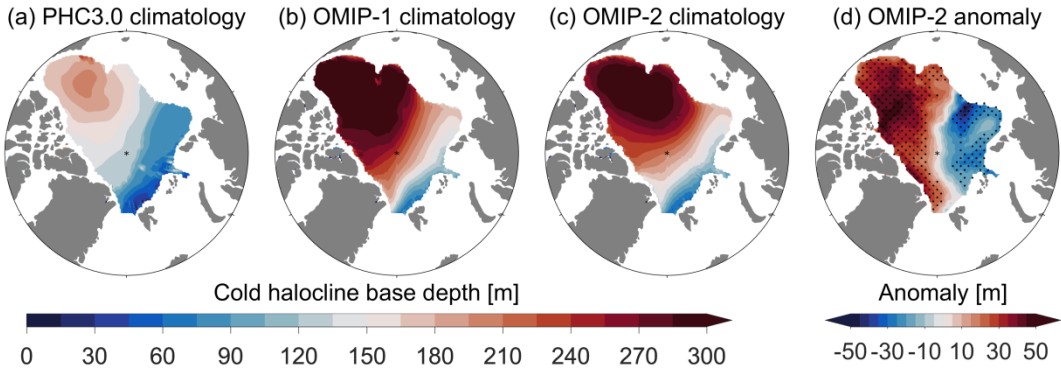

**Figure 11. Cold halocline base depth from (a) PHC3.0, (b) OMIP-1 and (c) OMIP-2 multi-model mean results. (d) Cold halocline base depth anomaly of 2009-2018 relative to 1979-2008 from OMIP-2. The average time period for (b) and (c) is 1971–2000. Dots in (d) indicate that the anomaly is larger than variability (one standard deviation of the results from 1979–2008). The eight models with bold model ID in Table 1 are used here.**

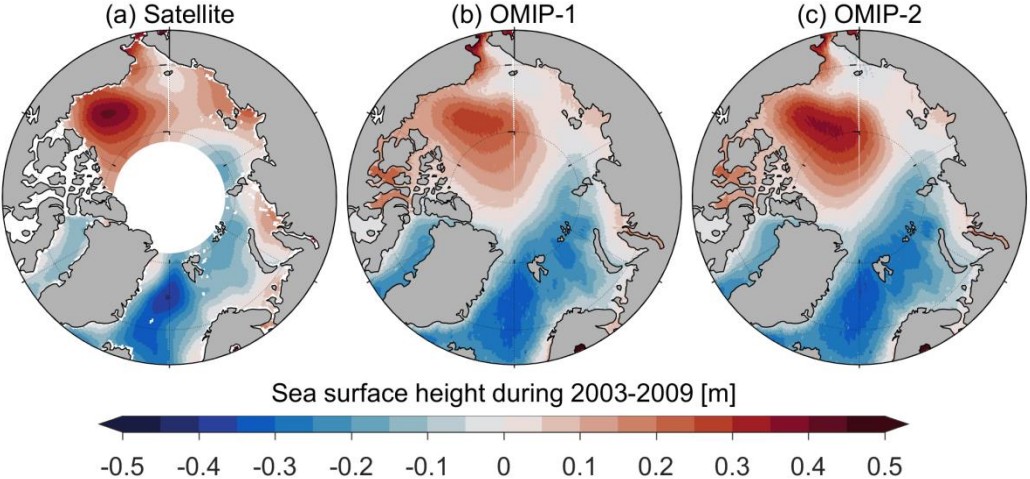

**Figure 12. Sea surface height from (a) satellite observations and the multi-model mean of (b) OMIP-1 and (c) OMIP-2 during 2003–2009.**

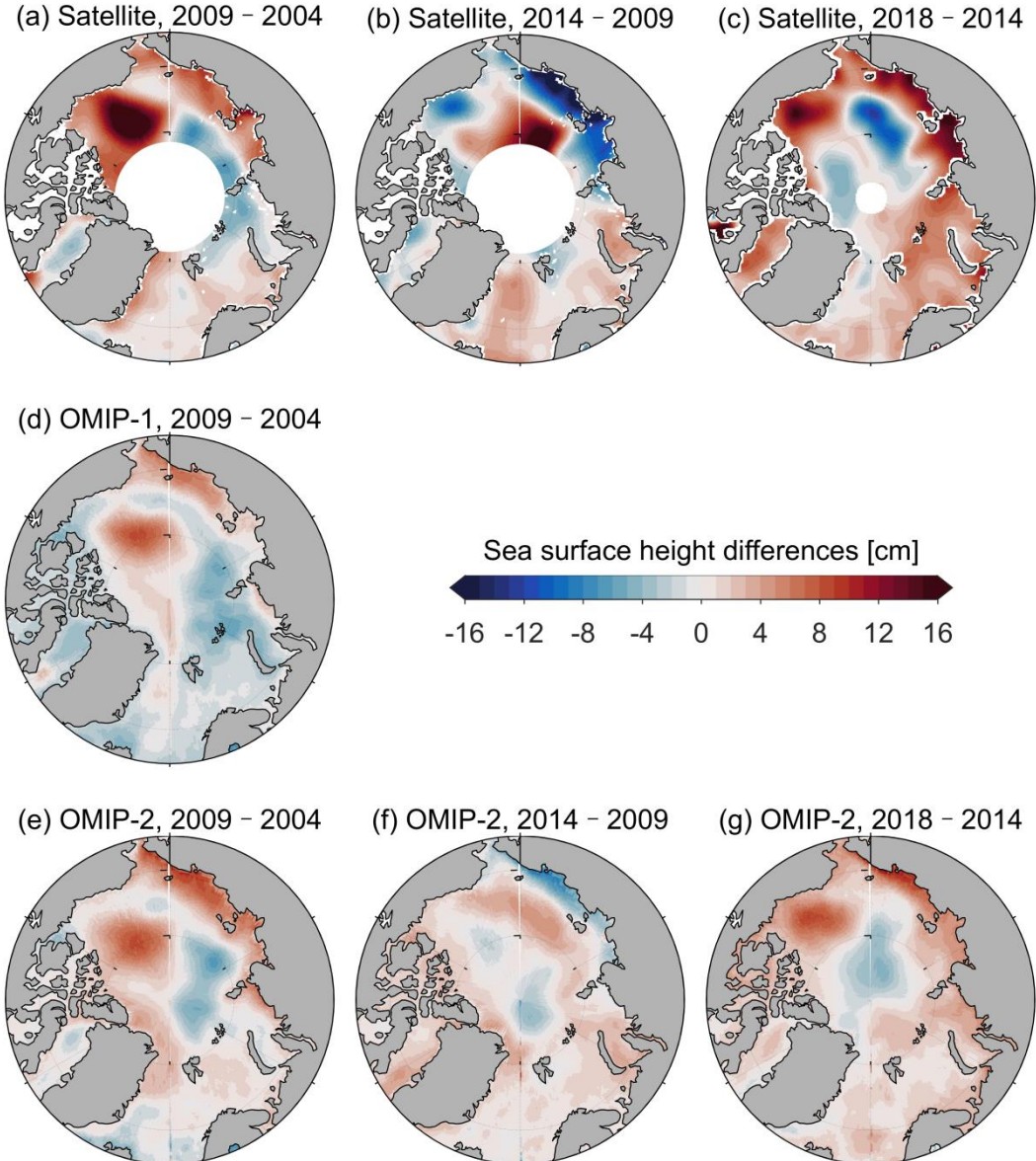

Figure 13. Sea surface height differences between (a) 2009 and 2004, (b) 2014 and 2009, and (c) 2018 and 2014 from satellite observations. (d) The same as (a), but for the OMIP-1 multi-model mean. (e)(f)(g) The same as (a)(b)(c), but for the OMIP-2 multi-model mean.

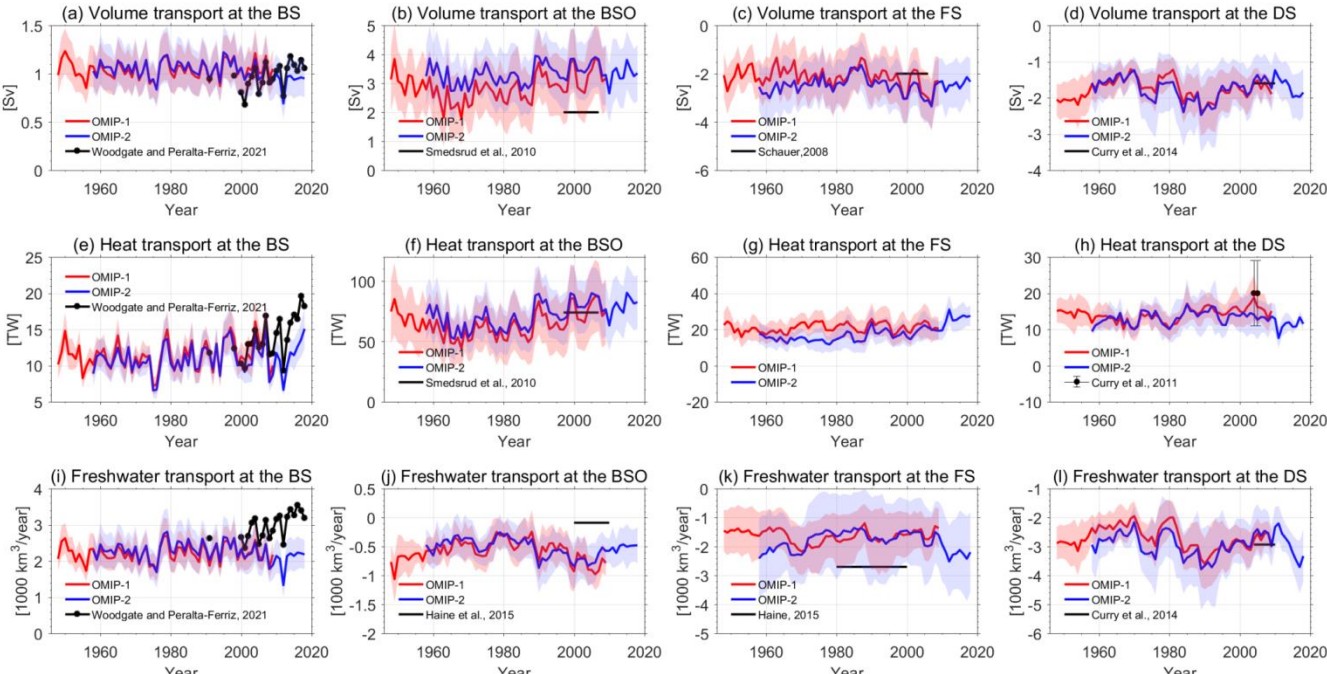

**Figure 14.** (upper) Volume, (middle) heat, and (bottom) liquid freshwater transports through the Bering Strait (BS), Barents Sea Opening (BSO), Fram Strait (FS), and Davis Strait (DS) in OMIP-1 (red), OMIP-2 (blue), and observations (black). The multi-model mean results are shown with lines, and the shading areas represent one standard deviation of the OMIP models. Seven models (AWI-CM-1-1-LR, CESM2, CMCC-CM2-SR5, EC-Earth3, MIROC6, MRI-ESM2-0, and NorESM2-LM) are used here. Ocean heat transport through the Bering Strait is calculated using reference temperature of −1.9 ℃ to be consistent with the observations. Reference temperature of 0 ℃ is used for other three gateways.

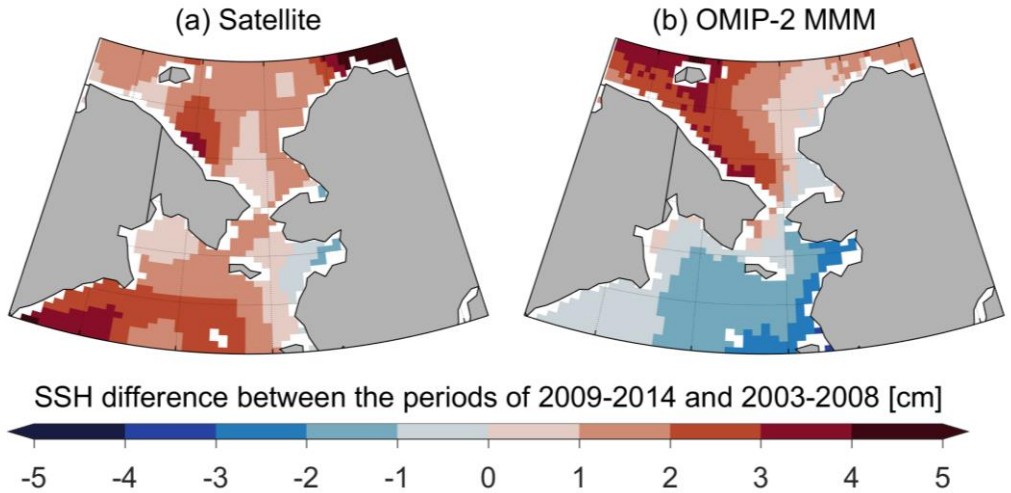

**Figure 15.** Sea surface height (SSH) difference between the period of 2009–2014 and the period of 2003–2008: (a) satellite observations, (b) OMIP-2 multi-model mean (MMM).

**Table 1. Model information. The eight models with bold model ID were used in both OMIP-1 and OMIP-2 simulations. Other models were only available in either OMIP-1 or OMIP-2 simulations.**

| No. | Model ID | Sea-ice Model | Ocean Model | Grid Number x×y×z | Vertical Coordinate |
|-----|----------|---------------|-------------|-------------------|---------------------|
| 1 | **AWI-CM-1-1-LR** | FESIM2 | FESOM1.4 | 126859[#]×46 | $z$ |
| 2 | CanESM5 | LIM2 | NEMO3.4.1 | 360×291×45 | $z$ |
| 3 | CanESM5-CanOE | LIM2 | NEMO3.4.1 | 360×291×45 | $z$ |
| 4 | CAS-ESM2-0 | CICE4 | LICOM2.0 | 360×196×30 | $\eta$ |
| 5 | **CESM2** | CICE5.1 | POP2 | 384×320×60 | $z$ |
| 6 | CMCC-CM2-HR4 | CICE4.0 | NEMO3.6 | 1442×1051×50 | $z$ |
| 7 | **CMCC-CM2-SR5** | CICE4.0 | NEMO3.6 | 360×291×50 | $z$ |
| 8 | CMCC-ESM2 | CICE4.0 | NEMO3.6 | 362×292×50 | $z$ |
| 9 | CNRM-CM6-1 | Gelato 6.1 | NEMO3.6 | 362×294×75 | $z$ |
| 10 | **EC-Earth3** | LIM3 | NEMO3.6 | 362×292×75 | $z$ |
| 11 | **FGOALS-f3-L** | CICE4.0 | LICOM3.0 | 360×218×30 | $\eta$ |
| 12 | FIO-ESM-2-0 | CICE4.0 | POP2-W | 384×320×61 | $z$ |
| 13 | GFDL-CM4 | GFDL-SIS2.0 | GFDL-MOM6 | 1440×1080×75 | hybrid $z-\rho(\sigma_2)$ |
| 14 | GFDL-OM4p5B | GFDL-SIS2.0 | GFDL-MOM6 | 720×576×75 | hybrid $z-\rho(\sigma_2)$ |
| 15 | IPSL-CM6A-LR | NEMO-LIM3 | NEMO-OPA | 362×332×75 | $z$ |
| 16 | **MIROC6** | COCO4.9 | COCO4.9 | 360×256×63 | hybrid $\sigma-z$ |
| 17 | MIROC-ES2L | COCO4.9 | COCO4.9 | 360×256×63 | hybrid $\sigma-z$ |
| 18 | **MRI-ESM2-0** | MRI.COM4.4 | MRI.COM4.4 | 360×363×61 | $z^*$ |
| 19 | **NorESM2-LM** | CICE5.1 | BLOM | 360×384×53 | $\rho(\sigma_2)$ |

[#] AWI-CM-1-1-LR employs unstructured meshes, and 126859 is the number of surface grid points.

**Table 2. Mean ocean volume transport and the standard deviation (unit: Sv) in the four Arctic Ocean gateways in OMIP-1. Positive values indicate flux into the Arctic Ocean. The results of the last cycle (1948–2009) are used in the analysis. Multi-model mean (MMM) is marked in bold.**

| Model | Bering Strait | | Barents Sea Opening | | Fram Strait | | Davis Strait | |
|---|---|---|---|---|---|---|---|---|
| | Mean | STD | Mean | STD | Mean | STD | Mean | STD |
| AWI-CM-1-1-LR | 0.8 | 0.1 | 2.0 | 0.4 | -2.0 | 0.4 | -1.0 | 0.2 |
| CAS-ESM2-0 | 1.1 | 0.1 | 0.3 | 0.3 | -1.3 | 0.3 | 0.0 | 0.0 |
| CESM2 | 0.8 | 0.1 | 1.4 | 0.5 | -0.7 | 0.5 | -1.6 | 0.1 |
| CMCC-CM2-SR5 | 1.1 | 0.1 | 3.9 | 0.6 | -2.9 | 0.7 | -2.0 | 0.5 |
| CMCC-ESM2 | 1.1 | 0.1 | 4.0 | 0.6 | -3.0 | 0.7 | -2.0 | 0.5 |
| CanESM5-CanOE | 0.7 | 0.1 | 4.1 | 0.5 | -3.0 | 0.5 | -1.6 | 0.3 |
| CanESM5 | 0.7 | 0.1 | 4.1 | 0.5 | -3.2 | 0.5 | -1.6 | 0.3 |
| EC-Earth3 | 1.3 | 0.1 | 4.2 | 0.6 | -2.6 | 0.5 | -2.3 | 0.5 |
| FIO-ESM-2-0 | 0.7 | 0.1 | 1.4 | 0.5 | -0.7 | 0.5 | -1.5 | 0.1 |
| IPSL-CM6A-LR | 1.3 | 0.1 | 3.9 | 0.7 | -2.6 | 0.8 | -2.4 | 0.4 |
| MIROC-ES2L | 1.1 | 0.1 | 3.6 | 0.4 | -3.8 | 0.4 | -1.5 | 0.2 |
| MIROC6 | 1.1 | 0.1 | 3.6 | 0.6 | -3.8 | 0.4 | -1.5 | 0.2 |
| MRI-ESM2-0 | 1.3 | 0.1 | 1.9 | 0.4 | -1.5 | 0.5 | -2.2 | 0.4 |
| NorESM2-LM | 0.8 | 0.1 | 2.9 | 0.5 | -1.5 | 0.4 | -1.8 | 0.3 |
| **MMM** | **1.0** | **0.1** | **3.0** | **0.5** | **-2.3** | **0.5** | **-1.6** | **0.3** |

**Table 3. Mean ocean volume transport and the standard deviation (unit: Sv) in the four Arctic Ocean gateways in OMIP-2. Positive values indicate flux into the Arctic Ocean. The results of the last cycle (1958–2018) are used in the analysis. Multi-model mean (MMM) is marked in bold.**

| Model | Bering Strait | | Barents Sea Opening | | Fram Strait | | Davis Strait | |
|---|---|---|---|---|---|---|---|---|
| | Mean | STD | Mean | STD | Mean | STD | Mean | STD |
| AWI-CM-1-1-LR | 0.9 | 0.1 | 2.5 | 0.4 | -2.9 | 0.3 | -0.6 | 0.2 |
| CESM2 | 0.7 | 0.1 | 2.0 | 0.4 | -1.3 | 0.3 | -1.5 | 0.2 |
| CMCC-CM2-HR4 | 1.2 | 0.2 | 3.5 | 0.3 | -2.1 | 0.5 | -2.2 | 0.4 |
| CMCC-CM2-SR5 | 1.2 | 0.2 | 4.0 | 0.4 | -2.4 | 0.7 | -2.6 | 0.7 |
| CNRM-CM6-1 | 1.3 | 0.1 | 4.6 | 0.5 | -4.3 | 0.4 | -1.7 | 0.5 |
| EC-Earth3 | 1.4 | 0.1 | 4.4 | 0.4 | -3.1 | 0.5 | -2.4 | 0.5 |
| MIROC6 | 1.2 | 0.1 | 4.4 | 0.4 | -3.9 | 0.3 | -1.4 | 0.2 |
| MRI-ESM2-0 | 1.3 | 0.1 | 2.0 | 0.3 | -1.8 | 0.4 | -1.8 | 0.3 |
| NorESM2-LM | 0.8 | 0.1 | 2.8 | 0.4 | -1.3 | 0.3 | -2.0 | 0.4 |
| **MMM** | **1.1** | **0.1** | **3.3** | **0.4** | **-2.6** | **0.4** | **-1.8** | **0.4** |

**Table 4. Mean ocean heat transport and the standard deviation (unit: TW) in the four Arctic Ocean gateways in OMIP-1. Positive values indicate flux into the Arctic Ocean. The results of the last cycle (1948–2009) are used in the analysis. Multi-model mean (MMM) is marked in bold.**

| Model | Bering Strait | | Barents Sea Opening | | Fram Strait | | Davis Strait | |
|---|---|---|---|---|---|---|---|---|
| | Mean | STD | Mean | STD | Mean | STD | Mean | STD |
| AWI-CM-1-1-LR | 2.3 | 1.6 | 62.3 | 8.2 | 24.0 | 7.7 | 15.8 | 5.4 |
| CAS-ESM2-0 | 7.5 | 1.8 | 18.6 | 3.7 | 25.3 | 3.2 | 6.3 | 1.1 |
| CESM2 | 3.6 | 1.4 | 30.2 | 8.1 | 14.1 | 2.8 | 14.6 | 1.4 |
| CMCC-CM2-SR5 | 3.2 | 2.0 | 84.4 | 13.7 | 22.6 | 4.1 | 16.6 | 4.6 |
| CMCC-ESM2 | 3.2 | 2.0 | 88.9 | 14.5 | 24.7 | 5.0 | 14.9 | 4.2 |
| CanESM5-CanOE | 3.4 | 1.4 | 84.1 | 13.5 | 17.3 | 2.4 | 8.9 | 1.4 |
| CanESM5 | 3.4 | 1.4 | 84.1 | 13.5 | 16.6 | 2.2 | 8.9 | 1.4 |
| EC-Earth3 | 2.9 | 2.0 | 90.4 | 15.0 | 12.7 | 3.3 | 11.0 | 2.0 |
| FIO-ESM-2-0 | 3.3 | 1.3 | 29.4 | 8.3 | 14.9 | 3.0 | 14.1 | 1.4 |
| IPSL-CM6A-LR | 2.6 | 2.0 | 83.3 | 16.6 | 20.8 | 6.5 | 13.3 | 2.3 |
| MIROC-ES2L | 3.5 | 1.6 | 86.3 | 12.7 | 29.8 | 3.1 | 11.8 | 2.3 |
| MIROC6 | 3.4 | 1.6 | 89.9 | 16.1 | 30.3 | 3.1 | 11.9 | 2.3 |
| MRI-ESM2-0 | 3.5 | 2.0 | 39.9 | 8.2 | 25.0 | 2.5 | 17.9 | 1.8 |
| NorESM2-LM | 2.7 | 1.1 | 55.5 | 10.5 | 18.3 | 3.2 | 10.8 | 2.0 |
| **MMM** | **3.5** | **1.7** | **66.2** | **11.6** | **21.2** | **3.7** | **12.6** | **2.4** |

**Table 5. Mean ocean heat transport and the standard deviation (unit: TW) in the four Arctic Ocean gateways in OMIP-2. Positive values indicate flux into the Arctic Ocean. The results of the last cycle (1958–2018) are used in the analysis. Multi-model mean (MMM) is marked in bold.**

| Model | Bering Strait | | Barents Sea Opening | | Fram Strait | | Davis Strait | |
|---|---|---|---|---|---|---|---|---|
| | Mean | STD | Mean | STD | Mean | STD | Mean | STD |
| AWI-CM-1-1-LR | 2.1 | 1.7 | 69.8 | 8.8 | 22.0 | 8.2 | 14.2 | 4.8 |
| CESM2 | 2.9 | 1.4 | 36.6 | 8.7 | 12.9 | 2.6 | 13.2 | 1.4 |
| CMCC-CM2-HR4 | 5.1 | 2.3 | 85.0 | 10.0 | 39.3 | 14.1 | 23.1 | 5.2 |
| CMCC-CM2-SR5 | 3.8 | 2.2 | 86.9 | 13.7 | 16.9 | 4.6 | 18.3 | 5.5 |
| CNRM-CM6-1 | 1.8 | 2.1 | 103.9 | 17.4 | 13.8 | 4.0 | 7.5 | 2.1 |
| EC-Earth3 | 2.7 | 2.1 | 98.1 | 13.6 | 13.0 | 5.8 | 10.7 | 2.0 |
| MIROC6 | 2.5 | 2.0 | 81.9 | 13.4 | 26.9 | 5.4 | 11.1 | 2.0 |
| MRI-ESM2-0 | 3.4 | 2.1 | 43.5 | 8.5 | 25.7 | 3.5 | 15.5 | 2.4 |
| NorESM2-LM | 2.6 | 1.3 | 59.1 | 10.7 | 14.8 | 5.6 | 10.3 | 2.7 |
| **MMM** | **3.0** | **1.9** | **73.9** | **11.6** | **20.6** | **6.0** | **13.8** | **3.1** |

**Table 6. Mean freshwater transport and the standard deviation (unit: $10^3$ km$^3$/year) in the four Arctic Ocean gateways in OMIP-1. Positive transports freshen the Arctic Ocean. The results of the last cycle (1948–2009) are used in the analysis. Multi-model mean (MMM) is marked in bold.**

| Model | Bering Strait | | Barents Sea Opening | | Fram Strait | | Davis Strait | |
|---|---|---|---|---|---|---|---|---|
| | Mean | STD | Mean | STD | Mean | STD | Mean | STD |
| AWI-CM-1-1-LR | 2.0 | 0.3 | -0.5 | 0.2 | -2.6 | 0.5 | -2.2 | 0.5 |
| CAS-ESM2-0 | 2.3 | 0.2 | -0.6 | 0.1 | -3.3 | 0.2 | -0.2 | 0.1 |
| CESM2 | 1.7 | 0.2 | -0.6 | 0.2 | -1.3 | 0.3 | -2.6 | 0.3 |
| CMCC-CM2-SR5 | 2.3 | 0.3 | -0.4 | 0.3 | -1.9 | 0.6 | -3.1 | 0.7 |
| CMCC-ESM2 | 2.4 | 0.3 | -0.3 | 0.3 | -2.2 | 0.7 | -3.2 | 0.7 |
| CanESM5-CanOE | 1.7 | 0.3 | -0.8 | 0.2 | -1.6 | 0.4 | -2.8 | 0.5 |
| CanESM5 | 1.7 | 0.3 | -0.8 | 0.2 | -1.5 | 0.4 | -2.8 | 0.5 |
| EC-Earth3 | 2.8 | 0.3 | -0.3 | 0.3 | -0.5 | 0.3 | -3.0 | 0.7 |
| FIO-ESM-2-0 | 1.5 | 0.2 | -0.5 | 0.1 | -1.4 | 0.3 | -2.6 | 0.2 |
| IPSL-CM6A-LR | 2.7 | 0.4 | -0.5 | 0.3 | -0.3 | 0.2 | -2.7 | 0.6 |
| MIROC-ES2L | 2.3 | 0.2 | -0.6 | 0.3 | -2.4 | 0.3 | -2.3 | 0.4 |
| MIROC6 | 2.3 | 0.2 | -0.5 | 0.3 | -2.4 | 0.2 | -2.3 | 0.4 |
| MRI-ESM2-0 | 2.6 | 0.3 | -0.6 | 0.1 | -2.1 | 0.3 | -3.8 | 0.6 |
| NorESM2-LM | 1.8 | 0.2 | -1.1 | 0.2 | -0.7 | 0.2 | -2.0 | 0.3 |
| **MMM** | **2.1** | **0.3** | **-0.6** | **0.2** | **-1.7** | **0.3** | **-2.5** | **0.5** |

**Table 7. Mean freshwater transport and the standard deviation (unit: $10^3$ km$^3$/year) in the four Arctic Ocean gateways in OMIP-2. Positive transports freshen the Arctic Ocean. The results of the last cycle (1958–2018) are used in the analysis. Multi-model mean (MMM) is marked in bold.**

| Model | Bering Strait | | Barents Sea Opening | | Fram Strait | | Davis Strait | |
|---|---|---|---|---|---|---|---|---|
| | Mean | STD | Mean | STD | Mean | STD | Mean | STD |
| AWI-CM-1-1-LR | 2.1 | 0.3 | -0.3 | 0.2 | -4.0 | 0.5 | -1.9 | 0.5 |
| CESM2 | 1.6 | 0.2 | -0.5 | 0.1 | -1.3 | 0.2 | -2.8 | 0.3 |
| CMCC-CM2-HR4 | 2.6 | 0.4 | -0.7 | 0.1 | -2.6 | 0.8 | -3.9 | 0.6 |
| CMCC-CM2-SR5 | 2.5 | 0.4 | -0.6 | 0.2 | -1.5 | 0.8 | -4.4 | 0.9 |
| CNRM-CM6-1 | 3.2 | 0.4 | 0.2 | 0.3 | -3.1 | 1.0 | -4.3 | 1.3 |
| EC-Earth3 | 3.1 | 0.4 | -0.1 | 0.2 | -0.7 | 0.4 | -3.4 | 0.6 |
| MIROC6 | 2.4 | 0.2 | -0.6 | 0.2 | -2.9 | 0.4 | -2.7 | 0.4 |
| MRI-ESM2-0 | 2.5 | 0.3 | -0.6 | 0.1 | -2.8 | 0.4 | -3.7 | 0.6 |
| NorESM2-LM | 1.8 | 0.2 | -1.1 | 0.2 | -0.5 | 0.3 | -2.4 | 0.4 |
| **MMM** | **2.4** | **0.3** | **-0.5** | **0.2** | **-2.2** | **0.5** | **-3.3** | **0.6** |