# Peer review of "Arctic Ocean Simulations in the CMIP6 Ocean Model Intercomparison Project (OMIP)"

_Geoscientific Model Development, 2022_

## Referee Comment (RC1)

**Review of "Arctic Ocean Simulations in the CMIP6 Ocean Model Intercomparison Project (OMIP)"**

This study compares the simulated Arctic Ocean across models in the latest OMIP experiments (OMIP-1 and OMIP-2 as designated in the manuscript) to those of previous CORE-II experiments by looking at several diagnostics including mean hydrography, liquid freshwater content, and transports through the major Arctic gateways. As a standard model intercomparison paper, the results are straightforward, and I recommend the manuscript for publication after the following concerns are addressed:

1. Methodology: The inter-model spread is used as a measure of the differences amongst the models and is defined as 1 standard deviation of a given value across the models. However, several of the models used in the OMIP have the same sea ice-ocean components (Table 1). How does this impact the spread and the multi-model mean when effectively some models are being double counted (i.e., those that use NEMO3.6 as the ocean model, or those that are MOM-based models under the hood)? It may not matter a ton here as I think the results are internally consistent by comparing across OMIPs, as long as the standard deviations aren't being used to make statistical inferences, but it might be good to clarify this a bit in the text if it's something that the authors thought about.

2. In a couple of locations in the manuscript it is noted that the AWI model performs best. Presumably, the 'best' model is defined is the one that is closest to observations? However, what exactly are the requirements for defining that especially when spatial variation is involved? The text could use a little more clarification regarding this. Another option is to add some difference maps between the models and observations in the supplementary material if that helps make the comparisons more obvious (I will leave that to the discretion of the authors though).

3. Citations: there are three other Arctic Ocean CMIP6 studies that could be cited in this manuscript in much the same way that the Wang et al. 2022b and Khosravi et al 2022 papers are invoked to connect the results of this study to its CMIP6 counterparts. Specifically, these studies warrant mention in the introduction and summary/conclusions (e.g., line 390) as well as in specific areas such as Sections 2.3 and 2.5.2 (Heuzé et al., *in review*), Section 2.4 (Muilwijk et al 2022), Sections 2.2, 2.5.1, and 2.5.3 (Zanowski et al., 2021, e.g., line 267—Zanowski noted the same volume transport trends in the Bering Strait as discussed here but in CMIP6 coupled models). References below:

   Heuzé, C.**,** H. Zanowski**,** S. Karam and M. Muilwijk: The deep Arctic Ocean and Fram Strait in CMIP6 models (*J. Climate*, in review). Preprint: https://eartharxiv.org/repository/view/3233/

Muilwijk, M., A. Nummelin, C. Heuzé, I. V. Polyakov, H. Zanowski, and L. H. Smedsrud: Divergence in climate model projections of future Arctic Ocean stratification and hydrography (J. Climate; https://doi.org/10.1175/JCLI-D-22-0349.1)

Zanowski, H.**,** A. Jahn, and M.M. Holland, 2021: Arctic Ocean freshwater in CMIP6 ensembles: Declining sea ice, increasing ocean storage and export, *JGR: Oceans,* **126**, doi.org/10.1029/2020JC016930

4.  Summary/Conclusions: The manuscript could benefit from further commentary about what it is we learned about the Arctic Ocean simulations from the OMIP comparisons that has not already been concluded in previous CMIP6 studies such as Khosravi et al 2022, Wang et al 2022b, Muilwijk et al 2022, Zanowski et al. 2021, and Heuzé et al. All of these studies note that simulation of the Arctic Ocean has not improved since CMIP5 (based on the diagnostics in those papers, of course). I am not doubting the validity or usefulness of the OMIPs or CORE simulations, but rather it would be helpful to place this study in the context of the other literature that has come out in the last year or two, and there is a nice opportunity to do that here by expanding this part of the manuscript.

5.  Lines 265-266 "The reasons for the discrepancy between observations and simulations are unknown and should be further investigated." Regarding the issues with the models being unable to reproduce the correct sign of the Bering Strait volume transport trend, didn't Wang et al 2022b (figure 8) suggest that this might be due to changes in the sea surface height gradient between the Arctic and Pacific? That may have been analysis for the future forcing scenarios and not the present-day trends as is the case in this manuscript, but it may also be worth commenting on. I have been wondering about the negative model volume transport trends vs. the positive trends observed by Woodgate et al. as well.

6.  A note on rainbow colormaps: It is best to avoid using a rainbow colormap where possible (there's a lot of information out there about why it's problematic). Please consider changing the colormaps for the figures to something non-rainbow. If any of the authors use python, cmocean has a nice set of pre-defined colormaps: https://matplotlib.org/cmocean/

---

## Author Comment (AC2)

**Response to Reviewer #2**

*(Note: Reviewer comments in black and our point-to-point replies in blue)*

**Comment on gmd-2022-260**

**Jonathan W. Rheinlaender (Referee)**

**Review of Arctic Ocean Simulations in the CMIP6 Ocean Model Intercomparison Project (OMIP)**

This paper analyses 19 ocean-sea ice models participating in the Ocean Model Intercomparison Project (OMIP) as part of CMIP6 and evaluates their performance in simulating Arctic Ocean properties. The simulations are evaluated mainly in terms of their mean climatological state, and to a lesser extent with respect to temporal variability.

The authors report that no significant improvements were made since the previous CORE II model simulations when it comes to simulating mean Arctic Ocean water mass and circulation properties. This is the main finding of the paper, with most models showing large biases in simulated mean hydrography.

The paper is nicely structured, well written and presents a valuable contribution to the modelling community. With some modifications it has potential to help guide future model development needed to improve ocean-sea ice models. Overall, I am pleased with the paper as is, but only have minor comments and suggestions. These are listed below along with some specific "in-text" comments.

Reply: We would like to express our thanks to the reviewer for all the valuable comments and suggestions, which helped to make the manuscript more solid and more useful for the ocean modelling community. We have tried to refine the manuscript following the reviewer's suggestions, and we believe that the manuscript is much improved by incorporating these comments.

**General Comments**

*Limited interpretation of the results*

The results are presented in a very "straightforward" manner which I presume is typical for GMD, but there is little interpretation and discussion of these results and various biases. I wish the authors would go beyond merely stating what the models are showing but interpret those results and put them into a larger context in the final discussion/conclusion section. For example, can the authors comment or speculate why there has been no improvements since CMIP5?

Reply: The reviewer raised a good point and asked a very valid question here. In the revised manuscript, we added some interpretation and discussion of these results and various biases.

For example, for the discrepancy in the trends of volume transport through the Bering Strait between observations and simulations, we investigated the changes in the sea surface height gradient between the Arctic and Pacific. We found that OMIP simulated changes in the sea surface height gradient between the Arctic and Pacific is much larger than the satellite observations, so it may lead to the discrepancy in the trends of volume transport through the Bering Strait between simulations and observations. In the revised manuscript, we added "The reasons for the discrepancy between observations and simulations may be caused by the larger changes in the OMIP simulated sea surface height gradient between the Arctic Ocean and Pacific compared to the observations (Fig. S16)".

[Figure]

Figure S16. (a) Observations and (b) OMIP-2 multi-model mean (MMM) sea surface height (SSH) difference between the period of 2009–2014 and the period of 2003–2008.

For the reasons why there have been no improvements since CMIP5, we add the following discussions in the discussion/conclusion section of the revised manuscript.

"Improving model parameterizations (e.g. horizontal and vertical mixing, brine rejection) and using higher model resolutions may be possible solutions. However, to our knowledge, targeted studies on improving parameterizations for the Arctic Ocean simulations have been very limited, if any at all since CORE-II, and the horizontal resolutions in most OMIP models are still coarse (nominal 1°, 24–50 km in the Arctic; Table 1). These factors may explain limited improvements from CORE-II to OMIP".

Furthermore, a paper like this offers an opportunity to reflect on the direction for the ocean modelling community going forward. Which model biases deserve the most attention? How do we fix them? Will increasing model resolution fix all these issues or do we need a different strategy? I would appreciate it if the authors could give their expert thoughts on this as I think it would be extremely useful to the community at large.

Reply: We appreciate these questions raised by the reviewer, which are all valid ones. In the revised manuscript, we added "According to the previous study (Shu et al., 2022), the Arctic Ocean warms faster than the global ocean mean in a warming climate, which is mainly contributed by the warming of the Atlantic Water layer. Considering the importance of the Atlantic Water in the Arctic Ocean climate change and the unrealistically deep and thick Atlantic Water layer simulated by ocean-sea ice and fully-coupled models, a high priority should be given to reduce the biases of Atlantic Water simulations in future model development. Wang et al. (2018) show that a model with 4.5-km resolution in the Arctic Ocean performs much better than a 24-km resolution model, especially in the simulations of the Atlantic Water layer, indicating that higher resolution (eddy permitting to eddy resolving) can help reduce model biases in Atlantic Water simulations through resolving eddy activity and reducing numerical mixing."

References

Shu, Q., Wang, Q., Årthun, M., Wang, S., and Song, Z.: Arctic Ocean Amplification in a warming climate in CMIP6 models, *Science Advances*, 1–11, 2022.

Wang, Q., Wekerle, C., Danilov, S., Wang, X., and Jung, T.: A 4.5g km resolution Arctic Ocean simulation with the global multi-resolution model FESOM 1.4, Geosci. Model Dev., 11, 1229–1255, https://doi.org/10.5194/gmd-11-1229-2018, 2018.

***No evaluation of horizontal circulation***

The paper focuses primarily on the model's capability of simulating Arctic Ocean hydrography and heat, volume and

freshwater exchanges. Is there a particular reason you do not evaluate the horizontal circulation? I suspect this could help in understanding some of the biases in water mass properties and exchanges across the different sections.

Reply: There are two reasons for which we did not evaluate the horizontal circulation. The first is that the CMIP6 models didn't provide information of angles between model grid directions and geographic directions, so we cannot get the horizontal velocity along geographic east and north for models not using lon-lat grids. The second is that we do not have effective observations regarding horizontal circulation to evaluate the models.

To evaluate Arctic upper ocean circulation, we added a section (Section 3.5) to compare the mean state and variability of modeled sea surface height with satellite observations in the manuscript, as follows:

**"3.5 Sea surface height**

Sea surface height in the Arctic basin area reflects the amount of liquid freshwater content (e.g., Morison et al., 2012; Wang, 2021). Furthermore, interannual changes in sea surface height are good indicators of interannual changes in the upper ocean circulation, because surface geostrophic currents dominate the Arctic surface velocity on spatial scales larger than 10 km and timescales longer than a few days (Doglioni et al., 2023). To evaluate the mean state and variability of upper ocean circulation, we compare modelled sea surface height with observations from altimetry measurements from Armitage et al. (2016).

The Arctic sea surface height is featured by a high in the Beaufort Sea associated with the anticyclonic Beaufort Gyre, a low in the Greenland Sea associated with the cyclonic Greenland Sea gyre and a large-scale gradient associated with the Transpolar Drift stream (Fig. 12a) (Armitage et al., 2016). Multi-model mean results of both OMIP-1 and OMIP-2 can reproduce the main features of the sea surface height in the Arctic (Fig. 12b and 12c). However, OMIP simulations have a broader and weaker Beaufort Gyre than the observed, and its centre is biased toward the Eurasian Basin. The Beaufort Gyre in OMIP-1 is weaker than that in OMIP-2 (Fig. 12), which is consistent with less liquid freshwater accumulated in the Beaufort Gyre in OMIP-1 than OMIP-2 (Fig. 7). The strength of the Beaufort Gyre and the location of its centre have large inter-model spread in OMIP models (Figs. S12 and S13). The multi-model mean cyclonic Greenland Sea gyre in both OMIP-1 and OMIP-2 is also weaker than the satellite observation (Fig. 12), with large inter-model spread (Figs. S12 and S13).

Major changes have occurred in the upper ocean circulation in the Arctic during the first two decades of the 21st century (Wang and Danilov, 2022), mainly featured by the unprecedented spin-up of the Beaufort Gyre. Satellite-derived sea surface height shows a marked spin-up of the Beaufort Gyre in the period of 2004–2009 (Fig. 13a), which was associated with the anomalous negative wind curl over the Canada Basin in this period. Satellite observations also show a reduction in the sea surface height between 2009 and 2004 in the Makarov and Eurasian basins, and a positive difference in the Laptev and East Siberian seas. In the period of 2009 to 2014, both the Beaufort High and Arctic Oscillation were close to neutral states on average, and a positive sea level pressure anomaly was centred over the outer shelf of the East Siberian Sea (Wang and Danilov, 2022). Consistently, satellite observations show a negative difference in the Beaufort, Laptev and East Siberian seas, and a positive difference in the Makarov Basin and over the outer shelf of the East Siberian Sea in this period (Fig. 13b). For the period of 2014–2018, negative wind curl anomalies in the southern part of the Canada Basin caused the spin-up of the Beaufort Gyre, which was confined to the southern Canada Basin in this period (Wang and Danilov, 2022; Fig. 13a and 13c). These changes in the sea surface height can be largely reproduced by the multi-model mean results of both OMIP-1 and OMIP-2 (Fig. 13d–g). However, the magnitude of the changes is underestimated by the multi-model mean, and the simulated changes in the Beaufort Gyre are also biased toward northwest. The negative difference in the Beaufort Sea between 2014 and 2009 is not reproduced by OMIP-2 multi-model mean (Fig. 13f)."

[Figure]

Figure 12. Sea surface height (m) from (a) satellite observations and the multi-model mean of (b) OMIP-1 and (c) OMIP-2 during 2003–2009.

[Figure]

Figure 13. Sea surface height differences between (a, d, e) 2009 and 2004, (b, f) 2014 and 2009, and (c, g) 2018 and 2014 from (a, b, c) satellite observations and the multi-model mean of (d) OMIP-1 and (e, f, g) OMIP-2.

[Figure]

Figure S12. Sea surface height (m) from satellite observation and OMIP-1 simulations during 2003–2009.

[Figure]

Figure S13. Sea surface height (m) from satellite observation and OMIP-2 simulations during 2003–2014.

The sea surface height gradient between the Arctic Ocean and Pacific mentioned above do help us to understand the trend bias of volume transport through the Bering Strait (Fig. S16).

*MMM versus individual model performance*

The paper focuses mostly on the MMM state, which I think is fine. But it would be nice if you could pull out specific models more. Why are some models performing well in certain cases? Why are some doing poorly? This could be an opportunity to learn what works and what does not. Specifically, I would like to see a more detailed discussion on the effects of model resolution (both horizontal and vertical) and choice of vertical coordinate.

Reply: We agree with the reviewer that more specific models should be discussed. We added more discussions about specific models.

For the effects of model resolution and choice of vertical coordinates, we added "These 19 ocean-sea ice models employ various horizontal and vertical resolutions and vertical coordinates (Table 1), however, no obvious grouping of models for model skills in terms of horizontal/vertical resolutions or vertical coordinates is found in this study. This may be partly caused by the fact that all the OMIP models evaluated here have very coarse resolutions. Furthermore, previous studies also found that greatly enhanced horizontal resolution does not deliver unambiguous bias improvement in all regions for all models (Chassignet et al.). We suggest that a dedicated high resolution Arctic Ocean model intercomparison project is needed".

**Specific Comments**

L64: "some systematic biases … have been identified". Could you list some examples?

Reply: This sentence was changed to "Significant progresses have been made in AOMIP and FAMOS, for example, some systematic biases in Arctic Ocean models (such as the progressive thickening and deepening of the AW layer) have been identified and some related solutions have been recommended".

L82: Consider making a new Methods section here.

Reply: Done.

L121: It would be helpful to the reader if you mention the typical resolution in the other models for comparison.

Reply: This sentence was changed to "The more realistic simulations in AWI-CM-1-1-LR may have benefited from its relatively high resolution (~24 km) in the Arctic Ocean, while the typical resolution in OMIP models is ~50 km".

L123-124: Can you please clarify why 400 m depth is chosen specifically? I guess, this is the depth of the AW layer, but this should be said explicitly. But I am wondering why you do not calculate and show the Ocean Heat Content for the AW layer? I think this could be a particularly useful diagnostic (for example for people interested in sea ice) and is easier to connect to the changes in ocean heat transport later. It would capture both biases related to the larger vertical extent of the AW layer and the temperature.

Reply: This sentence was changed to "Figure 3a–c shows the PHC3.0 and the multi-model mean potential temperature at 400 m, a depth that was used in previous model intercomparison studies (Ilicak et al., 2016), which is close to the core of the warm Atlantic Water layer in the Arctic Basin."

We agree with the reviewer that ocean heat content could be more useful than sea water temperature at a specific depth to evaluate AW layer simulations. However, according to the definition of the AW layer boundary, the thickness of AW layer in most OMIP models is much thicker than the observations, and there is no warm AW layer at all in some other models. Each model has its own drift, and it can be also challenging to define a common Atlantic layer boundary. So this will cause quite large uncertainty when we calculate ocean heat content for the AW layer. That is the main reason why we don't calculate the ocean heat content for the AW layer in the revised manuscript.

L164: It would be nice if you could motivate why the liquid freshwater content is an important metric to look at. Just one sentence at the start of the section.

Reply: "Liquid freshwater in the Arctic Ocean has strong implications on local physical and biogeochemical environment and downstream ocean circulations" was added in the revised manuscript.

L171-172: Can you comment on why OMIP-2 has more freshwater content in the Beaufort Sea compared to OMIP-1? Is there an improvement from CMIP5? Also, how important is sea ice for the freshwater biases in the models?

Reply: Tsujino et al. (2020) finds that the lower sea surface salinity in the Arctic Ocean in OMIP-2 relative to OMIP-1 is partly caused by the difference in salinity to which sea surface salinity is restored between OMIP-2 and OMIP-1. Sea ice decline has significant influences on the freshwater liquid freshwater accumulation in the Beaufort Gyre, by both supplying sea ice meltwater and by the convergence of other freshwater components (Wang et al., 2018). Tsujino et al. (2020) shows that sea ice volume in OMIP-2 has larger negative trend than OMIP-1, so more sea ice decline in OMIP-2 also contributes more freshwater in OMIP-2.

We added "The lower salinity in upper Arctic Ocean in OMIP-2 relative to OMIP-1 is partly caused by the difference in salinity to which sea surface salinity is restored between OMIP-2 (WOA13v2; Zweng et al., 2013) and OMIP-1 (PHC) (Tsujino et al., 2020). Sea ice decline also effects on the freshwater liquid freshwater accumulation in the Beaufort Gyre, by both supplying sea ice meltwater and convergence of other freshwater components (Wang et al., 2018). Tsujino et al. (2020) shows that sea ice volume in OMIP-2 has larger negative trend than OMIP-1, so more sea ice decline in OMIP-2 also partly contributes more liquid freshwater in OMIP-2" in the revised manuscript.

No significant improvements are found in the Arctic Ocean simulations from CMIP5 to CMIP6. "The similar biases are also reported in CMIP5 and CMIP6 fully-coupled models (Shu et al., 2018; Zanowski et al., 2021)" was added.

Reference:

Shu, Q., Qiao, F., Song, Z., Zhao, J., and Li, X.: Projected Freshening of the Arctic Ocean in the 21st Century, J. Geophys. Res. Ocean., 123, 9232–9244, https://doi.org/10.1029/2018JC014036, 2018.

Tsujino, H., Urakawa, L. S., Griffies, S. M., Danabasoglu, G., Adcroft, A. J., Amaral, A. E., Arsouze, T., Bentsen, M., Bernardello, R., Böning, C. W., Bozec, A., Chassignet, E. P., Danilov, S., Dussin, R., Exarchou, E., Fogli, P. G., Fox-Kemper, B., Guo, C., Ilicak, M., Iovino, D., Kim, W. M., Koldunov, N., Lapin, V., Li, Y., Lin, P., Lindsay, K., Liu, H., Long, M. C., Komuro, Y., Marsland, S. J., Masina, S., Nummelin, A., Rieck, J. K., Ruprich-Robert, Y., Scheinert, M., Sicardi, V., Sidorenko, D., Suzuki, T., Tatebe, H., Wang, Q., Yeager, S. G., and Yu, Z.: Evaluation of global ocean–sea-ice model simulations based on the experimental protocols of the Ocean Model Intercomparison Project phase 2 (OMIP-2), 3643–3708 pp., https://doi.org/10.5194/gmd-13-3643-2020, 2020.

Wang, Q., Wekerle, C., Danilov, S., Koldunov, N., Sidorenko, D., Sein, D., Rabe, B., and Jung, T.: Arctic Sea Ice Decline Significantly Contributed to the Unprecedented Liquid Freshwater Accumulation in the Beaufort Gyre of the Arctic Ocean, Geophys. Res. Lett., 45, 4956–4964, https://doi.org/10.1029/2018GL077901, 2018.

Zanowski, H., Jahn, A., and Holland, M. M.: Arctic Ocean Freshwater in CMIP6 Ensembles: Declining Sea Ice, Increasing Ocean Storage, and Export, J. Geophys. Res. Ocean., 126, 1–21, https://doi.org/10.1029/2020JC016930, 2021.

Zweng, M. ., Reagan, J. R., Antonov, J. I., Locarnini, R. A., Mishonov, A. V., Boyer, T. P., Garcia, H. E., Baranova, O. K., Johnson, D. R., Seidov, D., and Biddle, M. M.: NOAA Atlas NESDIS 74 WORLD OCEAN ATLAS 2013 Volume 2 : Salinity, World Ocean Atlas, 2, 182, 2013.

L198: AW warming over the whole Arctic basin?

Reply: Yes. Observations from (Polyakov et al., 2020) show that AW warming over the whole Arctic basin.

[Figure]

**[2006−2017]−[1981−1995]**

b−a

c

Figure. AW core temperatures difference between 2006–2017 and 1981–1995. This figure is from Polyakov et al. (2020).

Reference:

Polyakov, I. V., Alkire, M. B., Bluhm, B. A., Brown, K. A., Carmack, E. C., Chierici, M., Danielson, S. L., Ellingsen, I., Ershova, E. A., Gårdfeldt, K., Ingvaldsen, R. B., Pnyushkov, A. V., Slagstad, D., and Wassmann, P.: Borealization of the Arctic Ocean in Response to Anomalous Advection From Sub-Arctic Seas, Front. Mar. Sci., 7, 7–8, https://doi.org/10.3389/fmars.2020.00491, 2020

L207: I found the part about the re-initialization a bit difficult to follow. Could you describe this in a bit more detail? Perhaps in the Methods section. And can you quantify the impact of the re-initialization compared to the natural variability in the models?

Reply: In the Methods section, "Upon reaching the end of the year 2009 in OMIP-1 and 2018 in OMIP-2, the forcing is returned to 1948 in OMIP-1 and 1958 in OMIP-2" was added.

Figure 9c indicates that the re-initialization caused warming in 1960s in the Eurasian is larger than the natural variability. "The impact of the re-initialization is even larger than the natural variability" was added in the revised manuscript.

L229-230: Consider reiterating why the halocline layer is important. For example, by insulating the AW from the sea ice.

Reply: This sentence was changed to "The Arctic cold halocline layer is an important insulator between the warm Atlantic Water layer and the cold surface mixed layer and sea ice above".

L224: I would be interested in seeing the temporal changes in mixed layer depth. Do some of the models simulate episodic deep convection in the Arctic basin and what impacts could this have?

Reply: We added the linear trend of mixed layer depth during 1978 to 2018 based on OMIP-2 simulation in Fig. 10. "Figure 10c shows that cold season mixed layer depth has positive trends in most of the Arctic Ocean over the last 40 year based on OMIP-2 multi-model mean result, while it has negative trends in the Norwegian Sea, Baffin Bay, southern Barents Sea, and part of the Greenland Sea. The negative trends are mainly caused by less ocean surface heat release in these regions in a warming climate (Shu et al., 2021)" was added in the revised manuscript.

For deep convection simulation, we added "Some models (CMCC-CM2-HR4, CMCC-CM2-SR5, CNRM-CM6-1, and EC-Earth3) simulate episodic deep convection (maximum of mixed layer depth deeper than 200 m) in part the Eurasian Basin (Fig. S9), which might lead to oceanic heat from Atlantic Water layer release to mixed layer and melt sea in these models" in the revised manuscript.

[Figure]

Figure 10. Cold season (November–May) (a, b) mean mixed layer depth during 1979 to 2009 and (c) its linear trend during 1978 to 2018 based on multi-model mean results. The dots in (a) and (b) are observations during 1979 to 2012 (Peralta-Ferriz and Woodgate, 2015). The eight models with bold model ID in Table 1 are used here.

[Figure]

Figure S9. The maximum of monthly mixed layer depth during 2000 to 2018 in OMIP-2 models.

L245: It would be nice if you could discuss the broader implications of the simulated biases in halocline depth. Why is it important?

Reply: "So the potential roles of future shoaling of the halocline base in fully-coupled models (Shu et al., 2022) in the atmosphere-ocean-sea ice interactions may be weakened by the too deep biases of halocline base" was added in the section of Discussion and Conclusions.

L265: Are the trends negative since 1990 for all the OMIP model simulations? Also, can you please comment of the importance of horizontal resolution on simulating transport through the BS. Do models with higher resolution better capture the gateway?

Reply: Yes. All the OMIP simulations have negative trends since 1990. The biases are not related to model horizontal resolutions. Now we are coordinating a study to evaluate much higher resolution simulations forced by JRA55-do forcing, and they also have the biases of negative trends.

This sentence was changed to "Observations indicate that the volume transport has a positive trend (0.01 Sv/year) during 1990 to 2019 (Woodgate and Peralta-Ferriz, 2021), while the trends in all OMIP simulations are negative since 1990 (Figs. 12a, S11 and S12), and they are not related to model horizontal resolutions" in the revised manuscript.

L268: "historical observations" - please clarify over which time period specifically.

Reply: This sentence was changed to "The mean net volume transport through the BSO is ~2.0 Sv based on historical observations during 1997 to 2007 (Smedsrud et al., 2010, 2013)".

L360: Here I really miss a more detailed discussion on why there have been no major improvements in hydrography since CMIP5. It is a quite powerful statement, so it deserves more reflection. See also my general comment.

Reply: As our reply to the general comment, "Improving model parameterizations (e.g. horizontal and vertical mixing, brine rejection) and using higher model resolutions may be possible solutions. However, to our knowledge, targeted studies on improving parameterizations for the Arctic Ocean simulations have been very limited, if any at all since CORE-II, and the horizontal resolutions in most OMIP models are still coarse (nominal 1 °, 24–50 km in the Arctic; Table 1). These factors may explain limited improvements from CORE-II to OMIP" were added in the discussion/conclusion section.

L392-397: I am glad to see that you discuss the effect of resolution. However, this should be expanded upon in more detail. For example, how does resolution affect the model's capability in simulating the volume, heat and freshwater transport through narrow straits? And what about the effects of vertical resolution?

Reply: We agree with the reviewer that we should study the effect of resolution on more detail simulations in the Arctic Ocean. However, the horizontal resolutions in OMIP models are mostly coarse, and the inter-model differences are quite large even with similar resolution. Therefore, it is difficult to unambiguously determine whether the improvements come from inter-model differences or model resolutions.

   Now we are coordinating a new study on the effects of resolution, and five models provided their simulations with both low-resolution and high-resolution, and their high resolutions (nominal resolution of 0.1 ° or higher) are much higher than OMIP models. In this way we can exclude the influences from the inter-model differences to study the effects of high resolution. We will investigate the effects of resolution systematically in the new study. So we didn't discuss more detail in this revised manuscript.

L403-405: This last paragraph was not so clear. Consider reformulating.

Reply: The last paragraph was reformulated:

"We did not find significant improvement in simulating Arctic Ocean using JRA55-do forcing than using CORE2 forcing. However, the simulated variability and trends of freshwater content and gateways transports agree well between OMIP-1 and OMIP-2. CORE2 forcing only contains atmosphere forcing and runoff during 1948 to 2009, and has stopped updating. Therefore, JRA55-do forcing which has been updated to date is a good alternative to CORE2 forcing for studying recent changes in the Arctic Ocean. Part of the difference in the simulated temperature, salinity, and cold halocline base depth between OMIP-1 and OMIP-2 is caused by the design of the OMIP simulations. OMIP models run for no less than five cycles of the forcing periods to spin up. Upon reaching the end of the year 2009 in OMIP-1 and 2018 in OMIP-2, the forcing is returned to 1948 in OMIP-1 and 1958 in OMIP-2. If there is rapid climate change near the end of the forcing period, such as OMIP-2, repeating the full cycle of the atmosphere forcing back from 1958 can leave a large amount of Arctic Ocean heat and freshwater from the preceding simulation cycle to the following cycle. Our analysis suggests to only repeat the atmosphere forcing in the 20th century which has relatively weak climate change signal in the model spinup cycles."

**Figure comments:**

For all figures: I would suggest putting the name of the plotted variable in the colorbar legend (not only the caption) and with units. This makes it easier to quickly see what the figure is showing without having to read the caption first.

Reply: Done.

Fig 3: Consider also showing the temperature and salinity anomalies with respect to PHC3.0. Also, can you please comment on the absence of very cold and fresh water south of the Fram Strait seen in PHC3.0.

Reply: The temperature and salinity anomalies with respect to PHC3.0 are shown in the revised manuscript.

The absence of very cold and fresh water in the Greenland Sea is mainly caused by the coarse resolutions in OMIP. These biases are reduced markedly in higher-resolution simulations. For example, AWI-CM-1-1-LR and CMCC-CM2-HR4, which have relatively high resolution in OMIP, performance much better than the multi-model mean results (Fig. R1), and these biases can be further reduced by higher resolution (4.5 km) [see Figure 4 in Wang et al. (2018)].

For the absence of very cold and fresh water south of the Fram Strait, the sentence "the Atlantic Water layer in both the OMIP-1 and OMIP-2 multi-model mean results is too cold" was changed to "the Atlantic Water layer in both the OMIP-1 and OMIP-2 multi-model mean results is too cold in the Arctic Basin and too warm in the Greenland Sea and Norwegian Sea", "The warm biases in the Greenland Sea are small in relatively high resolution models, such as AWI-CM-1-1-LR and CMCC-CM2-HR4 (not shown)", and the sentence "In the Greenland Sea and Norwegian Sea, the biases of multi-model mean salinity are positive (Fig. 3i and 3j)" was added in the revised manuscript.

[Figure]

Figure R1. (upper) Potential temperature and (bottom) salinity at 400 m during 1971–2000 from (a, c) AWI-CM-1-1-LR and (b, d) CMCC-CM2-HR4.

Fig 4: Label name and unit on colorbar

Reply: Done.

Fig 7: Label name on colorbar

Reply: Done.

Fig 9: Add unit and name. It is difficult to see the lower values (blues) in the upper ocean. It would also be nice to show the

time series of AW ocean heat content here.

Reply: Unit and name were added in the revised manuscript. This figure is mainly used to evaluate the changes in Atlantic Water layer, and we the upper ocean is not our main focus here. As our reply to the general comment, there are quite large uncertainties when we calculate ocean heat content for the modeled Atlantic Water layer. So we didn't show the time series of Atlantic Water ocean heat content in the revised manuscript.

Fig 10: It is difficult to see the values >100 m. Can you extend the upper limit of the color scale so to better see MLD in the Barents Sea. Also, please clarify if it is the mean or max over the cold season?

Reply: Done. In the revised manuscript, Figure 10 was changed to:

[Figure]

Figure 10. Cold season (November–May) (a, b) mean mixed layer depth during 1979 to 2009 and (c) its linear trend during 1978 to 2018 based on multi-model mean results. The dots in (a) and (b) are observations during 1979 to 2012 (Peralta-Ferriz and Woodgate, 2015). The eight models with bold model ID in Table 1 are used here.

Table 1: It would be useful if you could list the type of vertical coordinate here too. The grid number for AWI-CM-1-1-LR is 126859 x 46 (x, y, z). Is this a typo?

Reply: Vertical coordinate was added in the revised manuscript. For AWI-CM-1-1-LR, which employs unstructured meshes, 126859 is the number of horizontal meshes. We added a note for that in the revised manuscript.

Table 2-7: Would it be possible to somehow highlight which models perform better relative to observations? Also, MMM in bold font, or double line before final row.

Reply: It is difficult for us to highlight which models perform better relative to observations, because the observed transports through the Arctic gateways usually have large uncertainties and cover different time periods. MMM is marked in bold font in the revised manuscript.

---

## Author Response (AR1)

**Response to Reviewer #1**

*(Note: Reviewer comments in black and our point-to-point replies in blue)*

**Review of "Arctic Ocean Simulations in the CMIP6 Ocean Model Intercomparison Project (OMIP)"**

This study compares the simulated Arctic Ocean across models in the latest OMIP experiments (OMIP-1 and OMIP-2 as designated in the manuscript) to those of previous CORE-II experiments by looking at several diagnostics including mean hydrography, liquid freshwater content, and transports through the major Arctic gateways. As a standard model intercomparison paper, the results are straightforward, and I recommend the manuscript for publication after the following concerns are addressed:

Reply: We are very grateful to the reviewer's comments and thoughtful suggestions. We made the revision accordingly, and we believe that the manuscript is much improved by taking into account these comments.

1. Methodology: The inter-model spread is used as a measure of the differences amongst the models and is defined as 1 standard deviation of a given value across the models. However, several of the models used in the OMIP have the same sea ice-ocean components (Table 1). How does this impact the spread and the multi-model mean when effectively some models are being double counted (i.e., those that use NEMO3.6 as the ocean model, or those that are MOM-based models under the hood)? It may not matter a ton here as I think the results are internally consistent by comparing across OMIPs, as long as the standard deviations aren't being used to make statistical inferences, but it might be good to clarify this a bit in the text if it's something that the authors thought about.

Reply: We totally agree with the reviewer that the number of models in different model families affects the results of spread and multi-model mean. For the comparison of OMIP-1 and OMIP-2 in Figures 3, 4, 6, 7, 8, 9, 10, 11, and 14, multi-model mean and inter-model spread are calculated based on the eight models with bold model ID in Table 1, and, coincidentally or not, they use different ocean components. But for the gateway transports in Tables 2-7, we used all model to calculate multi-model mean and inter-model spread, so this may affect the gateway transport results. We added some sentences in the revised manuscript to clarify this, including:

In Section 3.6.1: "Eight models (CanESM5, CanESM5-CanOE, CMCC-CM2-HR4, CMCC-CM2-SR5, CMCC-ESM2, CNRM-CM6-1, EC-Earth3, and IPSL-CM6A-LR) employing the NEMO ocean model produce relatively large net volume transport through the BSO compared with other models. As a result, the ensemble mean may be biased toward NEMO-family models".

In Section 3.6.2: "Consistent with the volume transport, the eight NEMO-family models simulate relatively large ocean heat transport through the BSO compared with other models. Similar behaviours are found in the CMIP6 fully-coupled climate models with NEMO as their ocean components in both historical simulations and future projections (Pan et al., 2023)".

In Section 3.6.3: "Tables 6 and 7 also indicate that the eight NEMO-family models simulate relatively large freshwater export through the Davis Strait compared with other models".

2. In a couple of locations in the manuscript it is noted that the AWI model performs best. Presumably, the 'best' model is defined is the one that is closest to observations? However, what exactly are the requirements for defining that especially when spatial variation is involved? The text could use a little more clarification regarding this. Another option is to add some difference maps between the models and observations in the supplementary material if that helps make the comparisons more obvious (I will leave that to the discretion of the authors though).

Reply: We found AWI-CM-1-1-LR in OMIP-1 performs well in the simulation of potential temperature profiles, and AWI-CM-1-1-LR in OMIP-2 performs well in the simulation of Arctic Ocean stratification (mixed layer depth and cold halocline base depth).

For the potential temperature profile and winter mixed layer depth simulations, Figures 2a, 2b, S7 and S8 show both simulations and observations together, so it should be easy to compare.

For the cold halocline base depth simulations, we added the root-mean-square error (RMSE) in the figure to make the comparisons more obvious.

[Figure]

Figure S11. Cold halocline base depth (unit: m) from PHC3.0 climatology and OMIP-2 models average over 1971 to 2000. The root-mean-square error (RMSE) averaged over the Arctic Ocean is labeled in each panel.

3. Citations: there are three other Arctic Ocean CMIP6 studies that could be cited in this manuscript in much the same way that the Wang et al. 2022b and Khosravi et al 2022 papers are invoked to connect the results of this study to its CMIP6 counterparts. Specifically, these studies warrant mention in the introduction and summary/conclusions (e.g., line 390) as well as in specific areas such as Sections 2.3 and 2.5.2 (Heuzé et al., in review), Section 2.4 (Muilwijk et al 2022), Sections 2.2, 2.5.1, and 2.5.3 (Zanowski et al., 2021, e.g., line 267—Zanowski noted the same volume transport trends in the Bering Strait as discussed here but in CMIP6 coupled models).

References below:

Heuzé, C., H. Zanowski, S. Karam and M. Muilwijk: The deep Arctic Ocean and Fram Strait in CMIP6 models (J. Climate, in review). Preprint: https://eartharxiv.org/repository/view/3233/

Muilwijk, M., A. Nummelin, C. Heuzé, I. V. Polyakov, H. Zanowski, and L. H. Smedsrud: Divergence in climate model projections of future Arctic Ocean stratification and hydrography (J. Climate; https://doi.org/10.1175/JCLI-D-22-0349.1)

Zanowski, H., A. Jahn, and M.M. Holland, 2021: Arctic Ocean freshwater in CMIP6 ensembles: Declining sea ice, increasing ocean storage and export, JGR: Oceans, 126, doi.org/10.1029/2020JC016930

Reply: These recent studies are indeed quite relevant to our manuscript. Thanks for the information. They are now cited in appropriate places in the revised manuscript.

In Section 3.1: "Cold biases in the Atlantic Water layer in the Arctic Basin are also found in CMIP6 fully-coupled climate models (Khosravi et al., 2022; Heuzé et al., 2023)".

In Section 3.2: "Positive freshwater content biases are also reported in CMIP5 and CMIP6 fully-coupled models (Shu et al., 2018; Zanowski et al., 2021; Wang et al., 2022b)".

In Section 3.4: "The positive anomaly of the simulated cold halocline base depth in the Amerasian Basin agrees well with the observed positive trend since 1970 (Muilwijk et al., 2022), which is not surprising since the models largely

reproduced the observed freshwater accumulation in the Arctic Basin (Fig. 8b)".

In Section 3.6.1: "The observed upward volume transport trend is also not reproduced in historical simulations of CMIP6 fully-coupled models (Zanowski et al., 2021, Wang et al., 2022b)".

In Section 3.6.2: "Most CMIP6 fully-coupled models also exhibit similarly low net ocean heat transport through the Fram Strait (Heuzé et al., 2023)".

In Section 4: "Therefore, it is not surprising that large biases and inter-model spread are found in the Arctic Ocean temperature and salinity simulations in CMIP6 fully-coupled models, and no significant improvements are found in the Arctic Ocean simulations from CMIP5 to CMIP6 (Khosravi et al., 2022; Wang et al., 2022b; Zanowski et al., 2021; Muilwijk et al., 2022; Heuzé et al., 2023)".

In Section 4: "OMIP models also have large inter-model spreads in the simulation of the Arctic Ocean stratification, which is similar to the performance of the CMIP6 fully-coupled models (Muilwijk et al., 2022).

4. Summary/Conclusions: The manuscript could benefit from further commentary about what it is we learned about the Arctic Ocean simulations from the OMIP comparisons that has not already been concluded in previous CMIP6 studies such as Khosravi et al 2022, Wang et al 2022b, Muilwijk et al 2022, Zanowski et al. 2021, and Heuzé et al. All of these studies note that simulation of the Arctic Ocean has not improved since CMIP5 (based on the diagnostics in those papers, of course). I am not doubting the validity or usefulness of the OMIPs or CORE simulations, but rather it would be helpful to place this study in the context of the other literature that has come out in the last year or two, and there is a nice opportunity to do that here by expanding this part of the manuscript.

Reply: We added some sentences in this section to connect the simulations from OMIP and CMIP6, including:

"The biases and inter-model spreads in OMIP are relatively small compared with those of CMIP6 fully-coupled models reported by Khosravi et al. (2022), which also indicates that part of biases of the Arctic Ocean simulations in the fully-coupled models is caused by the biases in the atmospheric component models (Hinrichs et al., 2021)".

"Therefore, it is not surprising that large biases and inter-model spread are found in the Arctic Ocean temperature and salinity simulations in CMIP6 fully-coupled models, and no significant improvements are found in the Arctic Ocean simulations from CMIP5 to CMIP6 (Khosravi et al., 2022; Wang et al., 2022b; Zanowski et al., 2021; Muilwijk et al., 2022; Heuzé et al., 2023)".

"Fully-coupled climate models have similar fresh biases in the halocline (Wang et al., 2022b), so the potential impacts of future shoaling of the halocline base in a warming climate (Shu et al., 2022) on the atmosphere-ocean-sea ice interactions may be biased low in climate model projections".

"OMIP models also have large inter-model spreads in the simulation of the Arctic Ocean stratification, which is similar to the performance of the CMIP6 fully-coupled models (Muilwijk et al., 2022)".

5. Lines 265-266 "The reasons for the discrepancy between observations and simulations are unknown and should be further investigated." Regarding the issues with the models being unable to reproduce the correct sign of the Bering Strait volume transport trend, didn't Wang et al 2022b (figure 8) suggest that this might be due to changes in the sea surface height gradient between the Arctic and Pacific? That may have been analysis for the future forcing scenarios and not the present-day trends as is the case in this manuscript, but it may also be worth commenting on. I have been wondering about the negative model volume transport trends vs. the positive trends observed by Woodgate et al. as well.

Reply: We agree with the reviewer that the changes in the sea surface height gradient between the Arctic and Pacific can cause the negative model volume transport. We compared OMIP-2 simulated sea surface height difference between the period of 2009–2014 and the period of 2003–2008 with the satellite observations. Figure 15 shows that simulated changes in the sea surface height gradient between the Arctic and Pacific is much larger than the satellite observations, so it may lead to

the discrepancy in the trends of volume transport through the Bering Strait between simulations and observations.

In the revised manuscript, this sentence was changed to "The OMIP models simulated a sea surface height drop throughout the northern Bering Sea in 2009–2014 relative to 2003–2008, with the largest decrease in the eastern Bering Sea (Fig. 15b). Satellite observations, on the contrary, revealed a sea surface height increase in most of the northern Bering Sea except for the Norton Sound for the same period (Fig. 15a). Furthermore, the sea surface height increase in the western Chukchi Sea is larger in the models than in the observation. The sea surface height gradient between the eastern Bering Sea and western Chukchi Sea is one important factor controlling the variability of the Bering Strait throughflow (Danielson et al., 2014; Peralta‑Ferriz and Woodgate, 2017; Zhang et al., 2020). Therefore, the reason for the discrepancy between the simulated and observed throughflow trends may be caused by the unrealistic reduction of sea surface height in the Bering Sea and the overestimated increase in the western Chukchi Sea as well in the 2010s in the OMIP-2 simulations. It remains to explore the reasons for the unrealistic representation of sea surface height changes in these regions in the future".

[Figure]

Figure 15. Sea surface height (SSH) difference between the period of 2009–2014 and the period of 2003–2008: (a) satellite observations, (b) OMIP-2 multi-model mean (MMM).

6. A note on rainbow colormaps: It is best to avoid using a rainbow colormap where possible (there's a lot of information out there about why it's problematic). Please consider changing the colormaps for the figures to something non-rainbow. If any of the authors use python, cmocean has a nice set of pre-defined colormaps:  https://matplotlib.org/cmocean/

Reply: Following the reviewer's suggestion, the colormaps from cmocean were used in the revised manuscript.

**Comment on gmd-2022-260**

**Jonathan W. Rheinlaender (Referee)**

**Review of Arctic Ocean Simulations in the CMIP6 Ocean Model Intercomparison Project (OMIP)**

This paper analyses 19 ocean-sea ice models participating in the Ocean Model Intercomparison Project (OMIP) as part of CMIP6 and evaluates their performance in simulating Arctic Ocean properties. The simulations are evaluated mainly in terms of their mean climatological state, and to a lesser extent with respect to temporal variability.

The authors report that no significant improvements were made since the previous CORE II model simulations when it comes to simulating mean Arctic Ocean water mass and circulation properties. This is the main finding of the paper, with most models showing large biases in simulated mean hydrography.

The paper is nicely structured, well written and presents a valuable contribution to the modelling community. With some modifications it has potential to help guide future model development needed to improve ocean-sea ice models. Overall, I am pleased with the paper as is, but only have minor comments and suggestions. These are listed below along with some specific "in-text" comments.

Reply: We would like to express our thanks to the reviewer for all the valuable comments and suggestions, which helped to make the manuscript more solid and more useful for the ocean modelling community. We have tried to refine the manuscript following the reviewer's suggestions, and we believe that the manuscript is much improved by incorporating these comments.

**General Comments**

*Limited interpretation of the results*

The results are presented in a very "straightforward" manner which I presume is typical for GMD, but there is little interpretation and discussion of these results and various biases. I wish the authors would go beyond merely stating what the models are showing but interpret those results and put them into a larger context in the final discussion/conclusion section. For example, can the authors comment or speculate why there has been no improvements since CMIP5?

Reply: The reviewer raised a good point and asked a very valid question here. In the revised manuscript, we added some interpretation and discussion of these results and various biases.

For example, for the discrepancy in the trends of volume transport through the Bering Strait between observations and simulations, we investigated the changes in the sea surface height gradient between the Arctic and Pacific. We found that OMIP simulated changes in the sea surface height gradient between the Arctic and Pacific is much larger than the satellite observations, so it may lead to the discrepancy in the trends of volume transport through the Bering Strait between simulations and observations. In the revised manuscript, we added "The OMIP models simulated a sea surface height drop throughout the northern Bering Sea in 2009–2014 relative to 2003–2008, with the largest decrease in the eastern Bering Sea (Fig. 15b). Satellite observations, on the contrary, revealed a sea surface height increase in most of the northern Bering Sea except for the Norton Sound for the same period (Fig. 15a). Furthermore, the sea surface height increase in the western Chukchi Sea is larger in the models than in the observation. The sea surface height gradient between the eastern Bering Sea and western Chukchi Sea is one important factor controlling the variability of the Bering Strait throughflow (Danielson et al., 2014; Peralta‑Ferriz and Woodgate, 2017; Zhang et al., 2020). Therefore, the reason for the discrepancy between the simulated and observed throughflow trends may be caused by the unrealistic reduction of sea surface height in the Bering Sea and the overestimated increase in the western Chukchi Sea as well in the 2010s in the OMIP-2 simulations".

[Figure]

Figure 15. Sea surface height (SSH) difference between the period of 2009–2014 and the period of 2003–2008: (a) satellite observations, (b) OMIP-2 multi-model mean (MMM).

For the reasons why there have been no improvements since CMIP5, we add the following discussions in the discussion/conclusion section of the revised manuscript.

"Improving model parameterizations (e.g. horizontal and vertical mixing) and using higher model resolutions may be possible solutions. However, to our knowledge, targeted studies on improving parameterizations for the Arctic Ocean simulations have been very limited, if any at all since the CORE-II project, and the horizontal resolutions in most OMIP models are still coarse (nominal 1°, 24–50 km in the Arctic; Table 1). These factors may explain the lack of improvements from CORE-II to OMIP".

Furthermore, a paper like this offers an opportunity to reflect on the direction for the ocean modelling community going forward. Which model biases deserve the most attention? How do we fix them? Will increasing model resolution fix all these issues or do we need a different strategy? I would appreciate it if the authors could give their expert thoughts on this as I think it would be extremely useful to the community at large.

Reply: We appreciate these questions raised by the reviewer, which are all valid ones. In the revised manuscript, we added "According to the previous study (Shu et al., 2022), the Arctic Ocean warms faster than the global ocean mean in a warming climate, which is mainly contributed by the warming of the Atlantic Water layer. Considering the importance of the Atlantic Water in the Arctic Ocean climate change and the unrealistically deep and thick Atlantic Water layer simulated by ocean-sea ice and fully-coupled models, a high priority should be given to reduce the biases of Atlantic Water simulations in future model development. Wang et al. (2018) show that a model with 4.5-km resolution in the Arctic Ocean performs much better than a 24-km resolution model, especially in the simulations of the Atlantic Water layer, indicating that higher resolution (eddy permitting to eddy resolving) can help reduce model biases in Atlantic Water simulations through resolving eddy activity and reducing numerical mixing."

References

Shu, Q., Wang, Q., Årthun, M., Wang, S., and Song, Z.: Arctic Ocean Amplification in a warming climate in CMIP6 models, *Science Advances*, 1–11, 2022.

Wang, Q., Wekerle, C., Danilov, S., Wang, X., and Jung, T.: A 4.5g km resolution Arctic Ocean simulation with the global multi-resolution model FESOM 1.4, Geosci. Model Dev., 11, 1229–1255, https://doi.org/10.5194/gmd-11-1229-2018, 2018.

***No evaluation of horizontal circulation***

The paper focuses primarily on the model's capability of simulating Arctic Ocean hydrography and heat, volume and freshwater exchanges. Is there a particular reason you do not evaluate the horizontal circulation? I suspect this could help in understanding some of the biases in water mass properties and exchanges across the different sections.

Reply: There are two reasons for which we did not evaluate the horizontal circulation. The first is that the CMIP6 models didn't provide information of angles between model grid directions and geographic directions, so we cannot get the horizontal velocity along geographic east and north for models not using lon-lat grids. The second is that we do not have effective observations regarding horizontal circulation to evaluate the models.

To evaluate Arctic upper ocean circulation, we added a section (Section 3.5) to compare the mean state and variability of modeled sea surface height with satellite observations in the manuscript, as follows:

**"3.5 Sea surface height**

Changes in sea surface height in the Arctic Basin reflect the variation of liquid freshwater content (e.g., Morison et al., 2012; Wang et al., 2021). Furthermore, interannual changes in sea surface height are good indicators of interannual changes in the upper Arctic Ocean circulation (Morison et al., 2021), because surface geostrophic currents dominate the Arctic surface velocity on spatial scales larger than 10 km and timescales longer than a few days (Doglioni et al., 2023). To evaluate the mean state and variability of upper ocean circulation, we compare modelled sea surface height with observational estimates from altimetry measurements provided by Armitage et al. (2016).

The Arctic sea surface height is featured with a high in the Beaufort Sea associated with the anticyclonic Beaufort Gyre, a low in the Greenland Sea associated with the cyclonic Greenland Sea gyre, and a large-scale gradient associated with the Transpolar Drift stream (Fig. 12a) (Armitage et al., 2016). The multi-model mean results of both OMIP-1 and OMIP-2 can reproduce these main features of the sea surface height in the Arctic (Fig. 12b and 12c). However, OMIP simulations have a broader and weaker Beaufort Gyre than the observed, and its center is biased toward the Eurasian Basin. The Beaufort Gyre in OMIP-1 is weaker than that in OMIP-2 (Fig. 12), which is consistent with lower freshwater column in the Beaufort Gyre in OMIP-1 than in OMIP-2 (Fig. 7). The strength of the Beaufort Gyre and the location of its center have large inter-model spreads in OMIP models (Figs. S12 and S13). The multi-model mean cyclonic Greenland Sea gyre in both OMIP-1 and OMIP-2 is also weaker than the satellite observation (Fig. 12), with large inter-model spreads as well (Figs. S12 and S13).

Major changes have occurred in the upper ocean circulation in the Arctic during the first two decades of the 21st century (Wang and Danilov, 2022), mainly manifested by the unprecedented spin-up of the Beaufort Gyre. Satellite-derived sea surface height shows a marked spin-up of the Beaufort Gyre in the period of 2004–2009 (Fig. 13a), which was associated with the anomalous negative wind curl over the Canada Basin in this period. Satellite observations also show a reduction in the sea surface height from 2004 to 2009 in the Makarov and Eurasian basins, and an increase in the Laptev and East Siberian seas. In the period of 2009 to 2014, both the Beaufort High and Arctic Oscillation were close to neutral states on average, and a positive sea level pressure anomaly was centered over the outer shelf of the East Siberian Sea (Wang and Danilov, 2022). Consistently, satellite observations show a reduction in the Beaufort, Laptev and East Siberian Seas, and an increase in the Makarov Basin and over the outer shelf of the East Siberian Sea in this period (Fig. 13b). For the period of 2014–2018, a negative wind curl anomaly in the southern part of the Canada Basin caused a spin-up of the Beaufort Gyre, which was confined to the southern Canada Basin in this period (Wang and Danilov, 2022; Fig. 13a and 13c). These changes in the sea surface height can be largely reproduced by the multi-model mean results of both OMIP-1 and OMIP-2 (Fig. 13d–g). However, the magnitude of the changes is underestimated by the multi-model mean, and the simulated changes in the Beaufort Gyre are also biased toward northwest. The observed sea surface height reduction in the Beaufort Sea from 2009 to 2014 is not reproduced by the OMIP-2 multi-model mean (Fig. 13f)."

[Figure]

Figure 12. Sea surface height from (a) satellite observations and the multi-model mean of (b) OMIP-1 and (c) OMIP-2 during 2003–2009.

[Figure]

Figure 13. Sea surface height differences between (a) 2009 and 2004, (b) 2014 and 2009, and (c) 2018 and 2014 from satellite observations. (d) The same as (a), but for the OMIP-1 multi-model mean. (e)(f)(g) The same as (a)(b)(c), but for the OMIP-2 multi-model mean.

[Figure]

Figure S12. Sea surface height (m) from satellite observation and OMIP-1 simulations during 2003–2009.

[Figure]

Figure S13. Sea surface height (m) from satellite observation and OMIP-2 simulations during 2003–2014.

The sea surface height gradient between the Arctic Ocean and Pacific mentioned above do help us to understand the trend bias of volume transport through the Bering Strait (Fig. 15).

*MMM versus individual model performance*

The paper focuses mostly on the MMM state, which I think is fine. But it would be nice if you could pull out specific models more. Why are some models performing well in certain cases? Why are some doing poorly? This could be an opportunity to learn what works and what does not. Specifically, I would like to see a more detailed discussion on the effects of model resolution (both horizontal and vertical) and choice of vertical coordinate.

Reply: We agree with the reviewer that more specific models should be discussed. We added more discussions about specific models.

For the effects of model resolution and choice of vertical coordinates, we added "These 19 ocean-sea ice models employ various horizontal and vertical resolutions and vertical coordinates (Table 1), but we did not find obvious grouping of models in terms of horizontal/vertical resolutions or vertical coordinates for model skills. This may be partly caused by the fact that the majority of the OMIP models evaluated here have very coarse resolutions (nominal 1°). Furthermore, previous studies also found that greatly enhanced horizontal resolution does not deliver unambiguous bias improvement in all regions for all models (Chassignet et al., 2020). We suggest that a dedicated high resolution Arctic Ocean model intercomparison project is needed".

**Specific Comments**

L64: "some systematic biases … have been identified". Could you list some examples?

Reply: This sentence was changed to "Significant progresses have been made in AOMIP and FAMOS, for example, some systematic biases in Arctic Ocean models (such as a wrong circulation direction and an overestimation of the thickness and depth of the Atlantic Water layer) have been identified and solutions to some of the issues have been recommended (Holloway et al., 2007; Golubeva and Platov, 2007; Zhang and Steele, 2007; Proshutinsky et al., 2007; Aksenov et al., 2016; Hu et al., 2019)".

L82: Consider making a new Methods section here.

Reply: Done.

L121: It would be helpful to the reader if you mention the typical resolution in the other models for comparison.

Reply: This sentence was changed to "The more realistic simulations in AWI-CM-1-1-LR may have benefited from its relatively high resolution (~24 km) in the Arctic Ocean, while the typical resolution of OMIP models is ~50 km".

L123-124: Can you please clarify why 400 m depth is chosen specifically? I guess, this is the depth of the AW layer, but this should be said explicitly. But I am wondering why you do not calculate and show the Ocean Heat Content for the AW layer? I think this could be a particularly useful diagnostic (for example for people interested in sea ice) and is easier to connect to the changes in ocean heat transport later. It would capture both biases related to the larger vertical extent of the AW layer and the temperature.

Reply: This sentence was changed to "Figure 3a–c shows the PHC3.0 and the multi-model mean potential temperature at 400 m, a depth that was used in previous model intercomparison studies (Ilicak et al., 2016) , which is close to the core of the warm Atlantic Water layer in the Arctic Basin."

There is no warm AW layer at all in some of the models, so it is hard to define the layer in this case. Furthermore, modellers also tend to plot temperature at a certain depth when first checking their new simulations. Our figure can be used as a reference for the status of the community models.

L164: It would be nice if you could motivate why the liquid freshwater content is an important metric to look at. Just one

sentence at the start of the section.

Reply: "Liquid freshwater in the Arctic Ocean has strong implications on Arctic physical and biogeochemical environment and large-scale ocean circulation in the North Atlantic (Coupel et al., 2015; Ardyna and Arrigo, 2020; Zhang et al., 2021)" was added in the revised manuscript.

L171-172: Can you comment on why OMIP-2 has more freshwater content in the Beaufort Sea compared to OMIP-1? Is there an improvement from CMIP5? Also, how important is sea ice for the freshwater biases in the models?

Reply: Tsujino et al. (2020) finds that the lower sea surface salinity in the Arctic Ocean in OMIP-2 relative to OMIP-1 is partly caused by the difference in salinity to which sea surface salinity is restored between OMIP-2 and OMIP-1. Sea ice decline has significant influences on the freshwater liquid freshwater accumulation in the Beaufort Gyre, by both supplying sea ice meltwater and by the convergence of other freshwater components (Wang et al., 2018). Tsujino et al. (2020) shows that sea ice volume in OMIP-2 has larger negative trend than OMIP-1, so more sea ice decline in OMIP-2 also contributes more freshwater in OMIP-2.

We added "Tsujino et al. (2020) suggested that the lower salinity in upper Arctic Ocean in OMIP-2 relative to OMIP-1 is partly caused by the difference in salinity to which sea surface salinity is restored between OMIP-1 and OMIP-2. Sea ice decline can increase liquid freshwater accumulation in the Beaufort Gyre, by both supplying sea ice meltwater and increasing convergence of other freshwater components (Wang et al., 2018b). Tsujino et al. (2020) shows that sea ice volume in OMIP-2 has a larger negative trend than OMIP-1, so the larger sea ice decline in OMIP-2 can also partly contribute to the higher freshwater content in OMIP-2" in the revised manuscript.

No significant improvements are found in the Arctic Ocean simulations from CMIP5 to CMIP6. "Positive freshwater content biases are also reported in CMIP5 and CMIP6 fully-coupled models (Shu et al., 2018; Zanowski et al., 2021; Wang et al., 2022b)" was added.

Reference:

Shu, Q., Qiao, F., Song, Z., Zhao, J., and Li, X.: Projected Freshening of the Arctic Ocean in the 21st Century, J. Geophys. Res. Ocean., 123, 9232–9244, https://doi.org/10.1029/2018JC014036, 2018.

Tsujino, H., Urakawa, L. S., Griffies, S. M., Danabasoglu, G., Adcroft, A. J., Amaral, A. E., Arsouze, T., Bentsen, M., Bernardello, R., Böning, C. W., Bozec, A., Chassignet, E. P., Danilov, S., Dussin, R., Exarchou, E., Fogli, P. G., Fox-Kemper, B., Guo, C., Ilicak, M., Iovino, D., Kim, W. M., Koldunov, N., Lapin, V., Li, Y., Lin, P., Lindsay, K., Liu, H., Long, M. C., Komuro, Y., Marsland, S. J., Masina, S., Nummelin, A., Rieck, J. K., Ruprich-Robert, Y., Scheinert, M., Sicardi, V., Sidorenko, D., Suzuki, T., Tatebe, H., Wang, Q., Yeager, S. G., and Yu, Z.: Evaluation of global ocean–sea-ice model simulations based on the experimental protocols of the Ocean Model Intercomparison Project phase 2 (OMIP-2), 3643–3708 pp., https://doi.org/10.5194/gmd-13-3643-2020, 2020.

Wang, Q., Wekerle, C., Danilov, S., Koldunov, N., Sidorenko, D., Sein, D., Rabe, B., and Jung, T.: Arctic Sea Ice Decline Significantly Contributed to the Unprecedented Liquid Freshwater Accumulation in the Beaufort Gyre of the Arctic Ocean, Geophys. Res. Lett., 45, 4956–4964, https://doi.org/10.1029/2018GL077901, 2018.

Zanowski, H., Jahn, A., and Holland, M. M.: Arctic Ocean Freshwater in CMIP6 Ensembles: Declining Sea Ice, Increasing Ocean Storage, and Export, J. Geophys. Res. Ocean., 126, 1–21, https://doi.org/10.1029/2020JC016930, 2021.

L198: AW warming over the whole Arctic basin?

Reply: Yes. Observations from (Polyakov et al., 2020) show that AW warming over the whole Arctic basin.

[Figure]

Figure. AW core temperatures difference between 2006–2017 and 1981–1995. This figure is from Polyakov et al. (2020).

Reference:

Polyakov, I. V., Alkire, M. B., Bluhm, B. A., Brown, K. A., Carmack, E. C., Chierici, M., Danielson, S. L., Ellingsen, I., Ershova, E. A., Gårdfeldt, K., Ingvaldsen, R. B., Pnyushkov, A. V., Slagstad, D., and Wassmann, P.: Borealization of the Arctic Ocean in Response to Anomalous Advection From Sub-Arctic Seas, Front. Mar. Sci., 7, 7–8, https://doi.org/10.3389/fmars.2020.00491, 2020

L207: I found the part about the re-initialization a bit difficult to follow. Could you describe this in a bit more detail? Perhaps in the Methods section. And can you quantify the impact of the re-initialization compared to the natural variability in the models?

Reply: In the Methods section, "Upon reaching the end of the year 2009 in OMIP-1 and 2018 in OMIP-2, the forcing is returned to 1948 in OMIP-1 and 1958 in OMIP-2" was added.

Figure 9c indicates that the re-initialization caused warming in 1960s in the Eurasian that is larger than the natural variability. "The impact of the re-initialization is even larger than the natural decadal variability (Fig. 9c)" was added in the revised manuscript.

L229-230: Consider reiterating why the halocline layer is important. For example, by insulating the AW from the sea ice.

Reply: This sentence was changed to "The Arctic cold halocline layer is an important insulator between the warm Atlantic Water layer and the cold surface mixed layer and sea ice above".

L224: I would be interested in seeing the temporal changes in mixed layer depth. Do some of the models simulate episodic deep convection in the Arctic basin and what impacts could this have?

Reply: We added the linear trend of mixed layer depth during 1978 to 2018 based on OMIP-2 simulation in Fig. 10. "Figure 10c shows that the cold season mixed layer depth has positive trends in most of the Arctic Ocean over the last 40 years in the OMIP-2 multi-model mean, except for the Norwegian Sea, Baffin Bay, southern Barents Sea, and part of the Greenland Sea where the trends are negative. The negative trends along the Atlantic Water pathway are mainly caused by less ocean surface heat release in a warming climate (Shu et al., 2021)" was added in the revised manuscript.

For deep convection simulation, we added "Some models (CMCC-CM2-HR4, CMCC-CM2-SR5, CNRM-CM6-1, and EC-Earth3) simulate episodic deep convection (maximum of mixed layer depth deeper than 200 m) in the Nansen Basin (Fig. S9). It might bring oceanic heat from the Atlantic Water layer to the mixed layer and reduce sea ice thickness in these models" in the revised manuscript.

[Figure]

Figure 10. Cold season (November–May) mean mixed layer depth during 1979 to 2009 based on multi-model mean results: (a) OMIP-1, (b) OMIP-2. (c) Linear trend of cold season mixed layer depth during 1978 to 2018 in OMIP-2. The dots in (a) and (b) are observations during 1979 to 2012 (Peralta-Ferriz and Woodgate, 2015). The eight models with bold model ID in Table 1 are used here.

[Figure]

Figure S9. The maximum of monthly mixed layer depth during 2000 to 2018 in OMIP-2 models.

L245: It would be nice if you could discuss the broader implications of the simulated biases in halocline depth. Why is it important?

Reply: "Fully-coupled climate models have similar fresh biases in the halocline (Wang et al., 2022b), so the potential impacts of future shoaling of the halocline base in a warming climate (Shu et al., 2022) on the atmosphere-ocean-sea ice interactions may be biased low in climate model projections" was added in the section of Discussion and Conclusions.

L265: Are the trends negative since 1990 for all the OMIP model simulations? Also, can you please comment of the importance of horizontal resolution on simulating transport through the BS. Do models with higher resolution better capture the gateway?

Reply: Yes. All the OMIP simulations have negative trends since 1990. The biases are not related to model horizontal resolutions. Now we are coordinating a study to evaluate much higher resolution simulations forced by JRA55-do forcing, and they also have the biases of negative trends.

This sentence was changed to "Observations indicate that the volume transport has a positive trend (0.01 Sv/year) during 1990 to 2019 (Woodgate and Peralta-Ferriz, 2021), while the trends in all OMIP simulations are negative since 1990 (Figs. 14a, S14 and S15). There is no evidence that the simulated erroneous trends are related to model horizontal resolutions" in the revised manuscript.

L268: "historical observations" - please clarify over which time period specifically.

Reply: This sentence was changed to "The mean net volume transport through the BSO is ~2.0 Sv based on historical observations during 1997 to 2007 (Smedsrud et al., 2010, 2013)".

L360: Here I really miss a more detailed discussion on why there have been no major improvements in hydrography since CMIP5. It is a quite powerful statement, so it deserves more reflection. See also my general comment.

Reply: As our reply to the general comment, "Improving model parameterizations (e.g. horizontal and vertical mixing) and using higher model resolutions may be possible solutions. However, to our knowledge, targeted studies on improving parameterizations for the Arctic Ocean simulations have been very limited, if any at all since the CORE-II project, and the horizontal resolutions in most OMIP models are still coarse (nominal $1°$, 24–50 km in the Arctic; Table 1). These factors may explain the lack of improvements from CORE-II to OMIP" were added in the discussion/conclusion section.

L392-397: I am glad to see that you discuss the effect of resolution. However, this should be expanded upon in more detail. For example, how does resolution affect the model's capability in simulating the volume, heat and freshwater transport through narrow straits? And what about the effects of vertical resolution?

Reply: We agree with the reviewer that we should study the effect of resolution on more detail simulations in the Arctic Ocean. However, the horizontal resolutions in OMIP models are mostly coarse, and the inter-model differences are quite large even with similar resolution. Therefore, it is difficult to unambiguously determine whether the improvements come from inter-model differences or model resolutions.

Now we are coordinating a new study on the effects of resolution, and five models provided their simulations with both low-resolution and high-resolution, and their high resolutions (nominal resolution of $0.1°$ or higher) are much higher than OMIP models. In this way we can exclude the influences from the inter-model differences to study the effects of high resolution. We will investigate the effects of resolution systematically in the new study. So we didn't discuss more detail in this revised manuscript.

L403-405: This last paragraph was not so clear. Consider reformulating.

Reply: The last paragraph was reformulated:

"We did not find significant improvement in simulating Arctic Ocean using JRA55-do forcing than using CORE2 forcing. However, the simulated variability and trends of freshwater content and gateways transports agree well between OMIP-1 and OMIP-2. The CORE2 forcing dataset only contains atmosphere forcing and runoff till 2009 without being further updated. Therefore, JRA55-do forcing which has been updated to date (till 2022) is a good alternative to CORE2 forcing for studying recent changes in the Arctic Ocean in a model intercomparison framework until the modelling community agrees on newer forcing dataset for future OMIP. Part of the difference in the simulated temperature, salinity, and cold halocline base depth between OMIP-1 and OMIP-2 is caused by the design of the OMIP simulations. OMIP models run for no less than five cycles of the forcing periods as model spinup. Upon reaching the end of the year 2009 in OMIP-1 and 2018 in OMIP-2, the forcing is returned to 1948 in OMIP-1 and 1958 in OMIP-2. As the ocean climate state near the end of the forcing period is very different from the climatology of the 20th century, such as in OMIP-2 (jumping from 2018 to 1958), repeating the full cycle of the atmosphere forcing can leave a large amount of Arctic Ocean heat and freshwater from the preceding simulation

cycle to the following cycle. Our analysis suggests to only repeat the atmosphere forcing of the 20th century in the model spinup cycles, which has relatively weak climate change signals."

**Figure comments:**

For all figures: I would suggest putting the name of the plotted variable in the colorbar legend (not only the caption) and with units. This makes it easier to quickly see what the figure is showing without having to read the caption first.

Reply: Done.

Fig 3: Consider also showing the temperature and salinity anomalies with respect to PHC3.0. Also, can you please comment on the absence of very cold and fresh water south of the Fram Strait seen in PHC3.0.

Reply: The temperature and salinity anomalies with respect to PHC3.0 are shown in the revised manuscript.

The absence of very cold and fresh water in the Greenland Sea is mainly caused by the coarse resolutions in OMIP. These biases are reduced markedly in higher-resolution simulations. For example, AWI-CM-1-1-LR and CMCC-CM2-HR4, which have relatively high resolution in OMIP, perform much better than the multi-model mean results (Fig. R1), and these biases can be further reduced with higher resolution (4.5 km) [see Figure 4 in Wang et al. (2018)].

For the absence of very cold and fresh water south of the Fram Strait, the sentence "the Atlantic Water layer in both the OMIP-1 and OMIP-2 multi-model mean results is too cold" was changed to "the Atlantic Water layer in both the OMIP-1 and OMIP-2 multi-model mean results is too cold in the Arctic Basin and too warm in the Greenland Sea and Norwegian Sea, "The warm biases in the Greenland Sea are smaller in relatively high resolution models, including AWI-CM-1-1-LR and CMCC-CM2-HR4 (not shown)", and the sentence "In the Greenland Sea and Norwegian Sea, the biases of multi-model mean salinity are positive (Fig. 3i and 3j)" was added in the revised manuscript.

[Figure]

Figure R1. (upper) Potential temperature and (bottom) salinity at 400 m during 1971–2000 from (a, c) AWI-CM-1-1-LR and (b, d) CMCC-CM2-HR4.

Fig 4: Label name and unit on colorbar

Reply: Done.

Fig 7: Label name on colorbar

Reply: Done.

Fig 9: Add unit and name. It is difficult to see the lower values (blues) in the upper ocean. It would also be nice to show the time series of AW ocean heat content here.

Reply: Unit and name were added in the revised manuscript. This figure is mainly used to evaluate the changes in Atlantic Water layer, and the upper ocean is not our main focus here. As our reply to the general comment, there are quite large uncertainties when we define the Atlantic Water layer range in some models. So we didn't show the time series of Atlantic Water ocean heat content in the revised manuscript.

Fig 10: It is difficult to see the values >100 m. Can you extend the upper limit of the color scale so to better see MLD in the Barents Sea. Also, please clarify if it is the mean or max over the cold season?

Reply: Done. In the revised manuscript, Figure 10 was changed to:

[Figure]

Figure 10. Cold season (November–May) mean mixed layer depth during 1979 to 2009 based on multi-model mean results: (a) OMIP-1, (b) OMIP-2. (c) Linear trend of cold season mixed layer depth during 1978 to 2018 in OMIP-2. The dots in (a) and (b) are observations during 1979 to 2012 (Peralta-Ferriz and Woodgate, 2015). The eight models with bold model ID in Table 1 are used here.

Table 1: It would be useful if you could list the type of vertical coordinate here too. The grid number for AWI-CM-1-1-LR is 126859 x 46 (x, y, z). Is this a typo?

Reply: Vertical coordinate was added in the revised manuscript. For AWI-CM-1-1-LR, which employs unstructured meshes, 126859 is the number of surface grid points. We added a note for that in the revised manuscript.

Table 2-7: Would it be possible to somehow highlight which models perform better relative to observations? Also, MMM in bold font, or double line before final row.

Reply: It is difficult for us to highlight which models perform better relative to observations, because the observed transports through the Arctic gateways usually have large uncertainties and cover different time periods. MMM is marked in bold font in the revised manuscript.

---

## Referee Report (RR1)

**Review of GMD-2022-260-version2**

Thank you for addressing my comments for the first version of the manuscript. I am pleased to see that you have picked out specific models that stand out and provided some interpretation of the model results. I find the 2nd version of the manuscript satisfactory and recommend it be published after a few minor adjustments. These are specified below.

**Comments**

**OMIP versus "standard" CMIP6 models**
For the reader who is unfamiliar with the OMIP initiative, it is perhaps not so clear what the difference is compared to the CMIP6. Perhaps you could clarify; are the OMIP models global or regional domain? How do they differ from the fully coupled models used in CMIP6? Just one-two sentences before the Methods.

L75: I presume you mean "have not been evaluated in a systematic/collective manner?" I would assume that some of the models have been evaluated individually before?

L115: Perhaps you can add a sentence mentioning that observations are also not without problems/biases and should not be taken as the complete truth.

L130: Using the potential temperature at 400m depth to evaluate the AW makes sense when comparing to previous studies. I would still consider looking at the maximum temperature within a depth interval (or 0-degree isotherm), which could account for differences in the depth of the AW layer between models. By choosing a fixed depth you may bias the results. At least, it is something you should check.

L173-175: You state that there are no improvements in the mean state in the MMM. This is somewhat depressing – so I wonder if there are any improvements in any individual model?

L183: "meter" —> "meters"

L212: It would be nice if you could include a small paragraph about the model's ability to represent stratification in the Beaufort Sea in particular; i.e., how well do the simulate Pacific Summer Water (PSW). This would be interesting to report on since the models seem to capture the volume through BS quite well.

L311-313: Are the biases in the BG due to incorrect atmospheric forcing or model physics? Please clarify.

L317: Can you please clarify if the freshwater transport include both liquid and solid freshwater contributions? What is the relative importance of solid/liquid transport?

L132: What about vertical resolution?

L354: "… to 2018 and the trend …" —>"…although the trend"

L358: Why is the spread in volume transport for the models so low compared to the observations?

L374: "The OMIP models obtained upward trends …" —> "The OMIP models also obtained positive trends"

L380: It's interesting that the models using NEMO seem to have some common biases. Can you comment on why this might be the case? Perhaps mixing schemes? Too strong AMOC preconditioning a stronger inflow across the Greenland-Scotland Ridge? Or representation of bathymetry in the inflow region (e.g Heuzé and Årthun 2019 10.1525/elementa.354)?

L465: "… implying that changes in the upper Arctic Ocean are captured in the models": In terms of what? Please specify. It's possible for the model to simulate the SSH correctly but still not capture the water mass structure accurately.

L495: Great! Thanks for suggesting this – I agree.

L502: I wonder if you really resolve eddies at 4.5 km outside the deep Arctic basin, e.g., on the shelf regions in the Barents and Kara Sea where you have Atlantic Water? Consider changing/adding the reference to Wang et al 2020 (https://doi.org/ 10.1029/2020GL088550) which uses a horizontal resolution of 1 km.

**Figures**

Fig 3+4: If possible I would change the colormap for a,b, c and f, g h to one that is not divergent. For example ''Thermal" for a,b, c and "Haline" for f, g,h. The divergent colormap implies differences/anomalies.

Figure 9, 10, 11, 13: Same as above. There are better colormaps available for displaying these metrics (see cmocean) and will help the communicate the figures to the reader more effectively. Note that people associate certain colormaps with specific things.

The comments on the figures are minor details, but I would still encourage the authors to consider updating the figures with more appropriate colormaps. I will leave it to the authors and the editor to make that decision.

---

## Author Response (AR2)

**Response to Reviewer #1**

*(Note: Reviewer comments in black and our point-to-point replies in blue)*

I would like to thank the authors for their thorough responses to my comments and I have no further suggestions at this time. There are some minor grammatical errors throughout the text that should be corrected moving forward, but I will leave it to the authors to take care of those in the final review.

Reply: We appreciate your positive feedbacks. Thanks for your help.

**Response to Reviewer #2**

*(Note: Reviewer comments in black and our point-to-point replies in blue)*

Thank you for addressing my comments for the first version of the manuscript. I am pleased to see that you have picked out specific models that stand out and provided some interpretation of the model results. I find the 2nd version of the manuscript satisfactory and recommend it be published after a few minor adjustments. These are specified below.

Reply: We appreciate your positive feedbacks and valuable suggestions. We made the revision according to your comments and suggestions.

**Comments**

**OMIP versus "standard" CMIP6 models**

For the reader who is unfamiliar with the OMIP initiative, it is perhaps not so clear what the difference is compared to the CMIP6. Perhaps you could clarify; are the OMIP models global or regional domain? How do they differ from the fully coupled models used in CMIP6? Just one-two sentences before the Methods.

Reply: Following the reviewer's suggestion, the sentence of "Ocean-sea ice models participating in the CMIP6 OMIP are driven by the specified common atmosphere forcing data sets" at the beginning of the Methods was changed to "Most of OMIP models were used as the ocean components of the CMIP6 fully-coupled models; different from the latter, global ocean-sea ice models participating in the CMIP6 OMIP are driven by the specified common atmosphere forcing data sets".

L75: I presume you mean "have not been evaluated in a systematic/collective manner?" I would assume that some of the models have been evaluated individually before?

Reply: "However, Arctic Ocean simulations by the OMIP ocean models, most of which were used as the ocean components of fully-coupled models in CMIP6, have not been evaluated" was changed to "However, Arctic Ocean simulations by the OMIP ocean models, most of which were used as the ocean components of fully-coupled models in CMIP6, have not been evaluated systematically in model intercomparison studies."

L115: Perhaps you can add a sentence mentioning that observations are also not without problems/biases and should not be taken as the complete truth.

Reply: "One has to keep in mind that although the datasets used to evaluate models are mostly based on observations, they also have biases and uncertainties" was added in the revised manuscript.

L130: Using the potential temperature at 400m depth to evaluate the AW makes sense when comparing to previous studies. I would still consider looking at the maximum temperature within a depth interval (or 0-degree isotherm), which could account for differences in the depth of the AW layer between models. By choosing a fixed depth you may bias the results. At least, it is something you should check.

Reply: Following the reviewer's suggestion, we checked the simulations of the Atlantic Water core temperature (the maximum temperature of the Atlantic Water at each grid location) by OMIP models. Figures R1 and 3 indicate that the model performances and patterns in the simulations of Atlantic Water core temperature and 400 m depth temperature are quite similar, and that the conclusions are not changed by the evaluation of the simulated Atlantic Water core temperature.

[Figure]

Figure R1. Atlantic Water core temperature (m) in (a) PHC3.0, and (b) OMIP-1 and (c) OMIP-2 multi-model mean results.

L173-175: You state that there are no improvements in the mean state in the MMM. This is somewhat depressing – so I wonder if there are any improvements in any individual model?

Reply: If we use a single metric to evaluate the model, we may find improvements in certain individual models – which are indeed addressed in the paper where relevant. However, no significant improvements are found in individual models for most metrics used in this study.

L183: "meter" —> "meters"

Reply: Done.

L212: It would be nice if you could include a small paragraph about the model's ability to represent stratification in the Beaufort Sea in particular; i.e., how well do the simulate Pacific Summer Water (PSW). This would be interesting to report on since the models seem to capture the volume through BS quite well.

Reply: We added "The discrepancy of mixed layer depth between simulations and observations in the

Chukchi and Beaufort Seas is relatively small. This may be partially attributed to the good performance in the representation of Pacific Water volume transport through the Bering Strait in OMIP models (see the results in Section 3.6.1)" in the revised manuscript.

L311-313: Are the biases in the BG due to incorrect atmospheric forcing or model physics? Please clarify.

Reply: At present, it is difficult to disentangle the two factors and pinpoint the exact reason for the bias in BG. This topic is beyond the scope of the present paper and warrants to be investigated in the future.

L317: Can you please clarify if the freshwater transport include both liquid and solid freshwater contributions? What is the relative importance of solid/liquid transport?

Reply: We only evaluate the liquid freshwater transport in this study. To clarify that, "and freshwater transport FWT through each Arctic Ocean gateway are calculated as follows" was changed to "and liquid freshwater transport FWT through each Arctic Ocean gateway are calculated as follows". Arctic solid freshwater transport mainly outflows through the Fram Strait. The estimation by Haine et al. (2015) indicates that the solid (2300 km$^3$/year) and liquid (2700 km$^3$/year) components of freshwater transport through the Fram Strait are comparable.

Refference:

Haine, T. W. N., Curry, B., Gerdes, R., Hansen, E., Karcher, M., Lee, C., Rudels, B., Spreen, G., de Steur, L., Stewart, K. D., and Woodgate, R.: Arctic freshwater export: Status, mechanisms, and prospects, Glob. Planet. Change, 125, 13–35, 2015.

L132: What about vertical resolution?

Reply: We also did not find relation to vertical resolution. "…horizontal or vertical resolutions…"

L354: "… to 2018 and the trend …" —>"…although the trend"

Reply: Done.

L358: Why is the spread in volume transport for the models so low compared to the observations?

Reply: The spreads of models and observations are calculated differently. The spread for simulations is one standard deviation of all simulations by OMIP models. The spread for observations rather indicates observation uncertainty, which is large due to very coarse mooring resolution.

L374: "The OMIP models obtained upward trends …" —> "The OMIP models also obtained positive trends"

Reply: Done.

L380: It's interesting that the models using NEMO seem to have some common biases. Can you comment on why this might be the case? Perhaps mixing schemes? Too strong AMOC preconditioning a stronger inflow across the Greenland-Scotland Ridge? Or representation of bathymetry in the inflow region (e.g Heuzé and Årthun 2019 10.1525/elementa.354)?

Reply: Thank you for the useful information. Tsujino et al. (2020) shows that the AMOC in NEMO-family models (such as CMCC-NEMO and Kiel-NEMO) is relatively weaker in OMIP simulations. We added "It might be related to the different vertical mixing scheme (TKE turbulent closure scheme) or the representation of bathymetry in NEMO-family models" in the revised manuscript. This is a topic that needs further research.

Reference:

Tsujino, H., Urakawa, L. S., Griffies, S. M., Danabasoglu, G., Adcroft, A. J., Amaral, A. E., Arsouze, T., Bentsen, M., Bernardello, R., Böning, C. W., Bozec, A., Chassignet, E. P., Danilov, S., Dussin, R., Exarchou, E., Fogli, P. G., Fox-Kemper, B., Guo, C., Ilicak, M., Iovino, D., Kim, W. M., Koldunov, N., Lapin, V., Li, Y., Lin, P., Lindsay, K., Liu, H., Long, M. C., Komuro, Y., Marsland, S. J., Masina, S., Nummelin, A., Rieck, J. K., Ruprich-Robert, Y., Scheinert, M., Sicardi, V., Sidorenko, D., Suzuki, T., Tatebe, H., Wang, Q., Yeager, S. G., and Yu, Z.: Evaluation of global ocean–sea-ice model simulations based on the experimental protocols of the Ocean Model Intercomparison Project phase 2 (OMIP-2), 3643–3708 pp., https://doi.org/10.5194/gmd-13-3643-2020, 2020.

L465: "… implying that changes in the upper Arctic Ocean are captured in the models": In terms of what? Please specify. It's possible for the model to simulate the SSH correctly but still not capture the water mass structure accurately.

Reply: We agree with the reviewer. "implying that changes in the upper Arctic Ocean are captured in the models" was changed to "implying that changes in the upper circulation of Arctic Ocean are captured in the models".

L495: Great! Thanks for suggesting this – I agree.

Reply: Thank you for your valuable suggestion in the first round of review.

L502: I wonder if you really resolve eddies at 4.5 km outside the deep Arctic basin, e.g., on the shelf regions in the Barents and Kara Sea where you have Atlantic Water? Consider changing/adding the reference to Wang et al 2020 (https://doi.org/ 10.1029/2020GL088550) which uses a horizontal resolution of 1 km.

Reply: We agree with the review that 4.5 km resolution cannot fully resolve eddies in the Arctic Basin. "Wang et al. (2018) show that a model with 4.5-km resolution in the Arctic Ocean performs much better than a 24-km resolution model" was changed to "Wang et al. (2018) show that a model

with 4.5-km (marginally eddy-permitting) resolution in the Arctic Ocean performs much better than a 24-km resolution model". We stay with the reference to Wang et al. (2018) here as detailed model evaluation was done in this paper.

**Figures**

Fig 3+4: If possible I would change the colormap for a,b, c and f, g h to one that is not divergent. For example ''Thermal" for a,b, c and "Haline" for f, g, h. The divergent colormap implies differences/anomalies.

Figure 9, 10, 11, 13: Same as above. There are better colormaps available for displaying these metrics (see cmocean) and will help the communicate the figures to the reader more effectively. Note that people associate certain colormaps with specific things.

The comments on the figures are minor details, but I would still encourage the authors to consider updating the figures with more appropriate colormaps. I will leave it to the authors and the editor to make that decision.

Reply: Following the reviewer's suggestion, we tried the colormaps of "Thermal" and "Haline", and we find that the original colormap "balance" is better to show the differences between different panels, such as Figures 3, 9 and 13. So we didn't change the colormaps in the revised manuscript.

[Figure]

Figure 3. (upper) Potential temperature and (bottom) salinity at 400 m from (a,f) PHC3.0, (b,g) OMIP-1, and (c,h) OMIP-2 multi-model mean results, and the biases of (d,e) potential temperature and (i,j) salinity of (d,i) OMIP-1 and (e,j) OMIP-2.

[Figure]

Figure 9. Hovmöller diagram of basin-mean potential temperature (unit: °C) for the Eurasian Basin (a, c) and Amerasian Basin (b,d) in OMIP-1 (a,b) and OMIP-2 (c,d).

[Figure]

Figure 13. Sea surface height differences between (a) 2009 and 2004, (b) 2014 and 2009, and (c) 2018 and 2014 from satellite observations. (d) The same as (a), but for the OMIP-1 multi-model mean. (e)(f)(g) The same as (a)(b)(c), but for the OMIP-2 multi-model mean.